# An evaluation of the OMI satellite instrument's ability to observe boundary layer ozone pollution across China: application to 2005-2017 ozone trends

Lu Shen[1], Daniel J. Jacob[1], Xiong Liu[2], Guanyu Huang[3], Ke Li[1], Hong Liao[4], Tao Wang[5]

[1]John A. Paulson School of Engineering and Applied Sciences, Harvard University, Cambridge, MA 02138, USA
[2]Harvard-Smithsonian Center for Astrophysics, Cambridge, Massachusetts 02138, USA
[3]Environmental & Health Sciences, Spelman College, Atlanta, Georgia 30314, USA
[4]School of Environmental Science and Engineering, Nanjing University of Information Science & Technology, Nanjing 210044, China
[5]Department of Civil and Environmental Engineering, The Hong Kong Polytechnic University, Hong Kong

*Correspondence to*: Lu Shen (lshen@fas.harvard.edu)

**Abstract.** Nadir-viewing satellite observations of tropospheric ozone in the UV have been shown to have some sensitivity to boundary layer ozone pollution episodes, but so far they have not yet been compared to surface ozone observations collected by large-scale monitoring networks.  Here we use 2013-2017 surface ozone data from the new China Ministry of Ecology and Environment (MEE) network of ~1000 sites, together with vertical profiles from ozonesondes and aircraft, to quantify
the ability of tropospheric ozone retrievals from the OMI satellite instrument and to detect boundary layer ozone pollution in China. We focus on summer when ozone pollution in China is most severe and when OMI has the strongest sensitivity. After subtracting the Pacific background, we find that the 2013-2017 mean OMI ozone enhancements over eastern China have strong spatial correlation with the corresponding multiyear means in the surface afternoon observations ($R = 0.73$), and that OMI can estimate these multiyear means in summer afternoon surface ozone with a precision of 8 ppb. The OMI data show
significantly higher values on observed surface ozone episode days (>82 ppb) than on non-episode days. Day-to-day correlations with surface ozone are much weaker due to OMI instrument noise, and are stronger for sites in southern China (<34$^{o}$N; $R = 0.3-0.6$) than in northern China ($R = 0.1-0.3$) because of weaker retrieval sensitivity and larger upper tropospheric variability in the north. Ozonesonde data show that much of the variability of OMI ozone over southern China in summer is driven by the boundary layer. Comparison of 2005-2009 and 2013-2017 OMI data indicates that mean summer
afternoon surface ozone in southern China (including urban and rural regions) has increased by 3.5±3.0  ppb over the 8-year period and that the number of episode days per summer has increased by 2.2±0.4 (as diagnosed by an extreme value model), generally consistent with the few long-term surface records. Ozone increases have been particularly large in the Yangtze River Delta and in the Hubei, Guangxi, and Hainan provinces.

## 1 Introduction

Ozone in surface air is harmful to public health (Bell et al., 2004). It is produced by photochemical oxidation of volatile organic compounds (VOCs) in the presence of nitrogen oxides ($NO_x \equiv NO + NO_2$). Both VOCs and $NO_x$ are emitted in large amounts in polluted regions by fuel combustion and industry. Ozone pollution is a particularly severe problem in China, where the air quality standard of 82 ppb (maximum 8-h daily average) is frequently exceeded (Wang et al., 2017). Observations in eastern China have reported increasing ozone trends of 1-3 ppb $a^{-1}$ over the past decade (Sun et al., 2016; Gao et al., 2017; Ma et al., 2017; Li et al., 2019). The surface observations were very sparse until 2013, when data from a national network of ~1000 sites operated by the China Ministry of Ecology and Environment (MEE) started to become available. Here we use the MEE network data to evaluate the ability of the space-based Ozone Monitoring Instrument (OMI) to observe ozone pollution in China, and we use the OMI data going back to 2005 to infer long-term ozone pollution trends.

OMI measures atmospheric ozone absorption by solar backscatter in the UV (270-365 nm) (Levelt et al., 2006). It follows a long lineage of UV satellite instruments (TOMS series starting in 1979, GOME series starting in 1995) directed primarily at monitoring the total ozone column. Retrieval of tropospheric ozone (only ~10% of the column) from these instruments has mostly been done in the past by subtracting independent satellite measurements of stratospheric ozone (Fishman et al., 1987; Ziemke et al., 2011) or using the convective cloud differential method (Ziemke et al., 1998, 2019). OMI has sufficiently fine spectral resolution to allow direct retrieval of tropospheric ozone, although the sensitivity decreases strongly toward the surface because of Rayleigh scattering (Liu et al., 2010). The direct retrieval typically provides one piece of information for the tropospheric ozone column weighted towards the middle troposphere (Zhang et al., 2010).

A number of previous studies have shown that satellite observations of ozone can detect boundary layer ozone pollution events (Fishman et al., 1987; Shim et al., 2009; Eremenko et al., 2008; Hayashida et al., 2008;), including for Chinese urban plumes (Kar et al., 2010; Hayashida et al., 2015; Gaudel et al., 2018; Dufour et al., 2018). Even if sensitivity to the boundary layer is low, the enhancements can be sufficiently large to enable detection. However, no quantitative comparison of the satellite data to surface observations has so far been done. Surface ozone network data are available in the US and Europe but levels are generally too low to enable effective comparison. Ozone levels in China are much higher (Lu et al., 2018). The high density of the MEE network, combined with vertical profile information from ozonesondes and aircraft, provides a unique opportunity for evaluating quantitatively the ability of OMI to observe ozone pollution.

## 2 Data and Methods

We use the OMI Ozone Profile retrieval (PROFOZ v0.9.3, level 2) product (Liu et al., 2010; Kim et al., 2013; Huang et al.,

2017, 2018) from the Smithsonian Astrophysical Observatory (SAO). OMI is in polar sun-synchronous orbit with a 1330 local observation time, and provides daily global mapping with $13\times24$ km$^2$ nadir pixel resolution (Levelt et al., 2006). Partial ozone columns are retrieved by PROFOZ for 24 vertical layers, of which 3-7 are in the troposphere with pressure levels dependent on tropopause and surface pressure (Liu et al., 2010). The retrieval uses a Bayesian optimization algorithm with prior information from the McPeters et al. (2007) climatology varying only by latitude and month. Averaging kernel matrices quantifying retrieval sensitivity are provided for individual retrievals. The trace of the averaging kernel matrix below a given retrieval pressure (degrees of freedom for signal or DOFS) estimates the number of independent pieces of information on the ozone profile below that pressure. The DOFS for the tropospheric ozone column in summer as retrieved by PROFOZ is about 1 (Zhang et al., 2010). The PROFOZ tropospheric retrievals have been successfully validated with ozonesonde data (Huang et al., 2017).

We focus on summer when ozone pollution in China is most severe and when OMI has the strongest sensitivity (Zhang et al., 2010). Since 2009, certain cross-track OMI observations have degraded because of the so-called row anomaly (Kroon et al., 2011; Huang et al., 2017, 2018). We only use pixels that (1) pass the reported quality checks, (2) have a cloud fraction less than 0.3, and (3) have a solar zenith angle less than 60°.

The DOFS below 400 hPa over eastern China are in the range 0.3-0.6 (Figure 1a). The DOFS are higher in the south than in the north due to higher solar elevation in the south, and higher over China than in background air at the same latitude due to higher ozone abundances. We use DOFS > 0.3 in Figure 1a as criterion for further analysis; this excludes northern and western China. Even though a DOFS of 0.3 is still low, it is based on the prior estimate of low boundary layer ozone in the McPeters et al. (2007) zonal mean climatology. As we will see, the retrieval is sensitive to ozone enhancements in the boundary layer when these are sufficiently high.

The prior estimate from McPeters et al. (2007) includes a latitudinal gradient of ozone concentrations that may be retained in the retrieval. To remove this background gradient and also any long-term uniform drift in the data, we subtract the monthly mean Pacific background (150°E-150°W) from the OMI data over China for the corresponding latitude and month. The residual defines an OMI enhancement over China that we use for further analysis. This subtraction requires that we use a common pressure range for the OMI observations over China and the Pacific, but the OMI retrievals have variable pressure ranges depending on the local tropopause and surface pressure (Liu et al., 2010). The three lowest layers in the retrieval (L24-L22) have pressure ranges of approximately 1000-700 hPa, 700-500 hPa, and 500-350 hPa for a column based at sea level, and all contain some information on boundary layer ozone (Figure S1). Here we choose the pressure range 850-400 hPa to define the OMI enhancement relative to the Pacific background, and compute OMI columns for that pressure range by weighting the local L24-22 retrievals. Using 850 hPa as a bottom pressure avoids complications from variable topography in eastern China. The 850-400 hPa retrievals capture all of the OMI sensitivity below 850 hPa in any case. We examined

different spatial and temporal averaging domains for the North Pacific background and found little effect on the residual.

We compare the OMI ozone enhancements to ozone measurements from surface sites, ozonesondes, and aircraft. We use surface ozone measurements from the MEE network available for 2013-2017 (http://datacenter.mep.gov.cn/index). We select the summer (JJA) data at 12-15 local solar time (LT), corresponding to the OMI overpass. The network had 450 sites in 2013 and 1500 sites as of 2017, most located in large cities. We also use 2005-2016 summertime ozonesonde data at 12-15 LT for Hanoi (105.8°E, 21.0°N), Hong Kong (114.2°E, 22.3°N), Naha (127.7°E, 26.2°N), Tsukuba (140.1°E, 36.1°N), and Sapporo (141.3°E, 43.1°N), , available from the World Ozone and Ultraviolet Radiation Data Center (WOUDC) (http://woudc.org/). We further use take-off/landing vertical profiles at 12-15 LT over East Asia from the In-Service Aircraft for the Global Observing System (IAGOS, http://www.iagos-data.fr/). For evaluating the long-term surface ozone trends inferred from OMI, we use 2005-2014 trend statistics for maximum daily 8-hour average (MDA8) ozone from the Tropospheric Ozone Assessment Report (TOAR) (Schultz et al., 2017). We also have 2005-2017 JJA 12-15 LT mean ozone at the Hok Tsui station in Hong Kong (Wang et al., 2009).

### 3 Inference of surface ozone from OMI observations

Figure 1b shows the mean midday (12-15 LT) surface ozone for the summers of 2013-2017 as measured by the MEE network. Concentrations exceed 70 ppb over most of the North China Plain with particularly high values in the Beijing-Tianjin-Hebei (BTH) megacity cluster. Values are also high in the Yangtze River Delta (YRD), Pearl River Delta (PRD), Sichuan Basin (SCB), and the city of Wuhan in central China. High values extend to the region west of the North China Plain, which is less densely populated but has elevated terrain.

OMI mean ozone abundances at 850-400 hPa for the summers of 2013-2017 are shown in Figure 1c. Values are partial column concentrations in Dobson units (1 DU = $2.69 \times 10^{16}$ molecules cm$^{-2}$). After subtracting the North Pacific background for the corresponding latitude in month, we obtain the OMI ozone enhancements shown in Figure 1d. The spatial correlation coefficient between the OMI ozone enhancements and the MEE surface network is $R = 0.73$ over eastern China. The correlation is driven in part by the latitudinal gradient but also by the enhancements in the large megacity clusters identified as rectangles in Figure 1b. Thus the correlation coefficient is $R = 0.55$ for the 26-34°N latitude band including YRD, SCB, and Wuhan. Figure 1e shows the corresponding scatterplot and the reduced major axis (RMA) regression relating the OMI enhancement $\Delta\Omega$ to the 12-15 LT surface concentration [O$_3$] (the slope is 0.14 DU/ppb). From there one can estimate multi-year average surface ozone (ppb) on the basis of the observed OMI enhancement (DU) as

$$[O_3] = 6.9 \, \Delta\Omega + 24.6 \pm 8.4 \tag{1}$$

where the error standard deviation (precision) of 8 ppb is inferred from the scatterplot. With such a precision, OMI can

provide useful information on mean summer afternoon levels of surface ozone in polluted regions.

Capturing the day-to-day variability of surface ozone leading to high-ozone pollution episodes is far more challenging because of noise in individual retrievals. Figure 1f shows the OMI vs. MEE temporal correlation for the daily data. Correlation coefficients are consistently positive and statistically significant, but relatively weak. They are higher in southern China ($R = 0.3$-$0.6$) than in northern China ($R = 0.1$-$0.3$), consistent with the pattern of OMI information content (DOFS) in Figure 1a. This implies that OMI can only provide statistical rather than deterministic temporal information on ozone pollution episodes, and may be more useful in South than in North China. We return to this point in Section 4.

Figure 2 (top panel) shows the relationship of OMI enhancements with daily MEE surface ozone concentrations averaged spatially in each of the five megacity clusters identified in Figure 1. Consistent with the distribution of DOFS (Figure 1a), the correlations are higher in PRD, SCB, and Wuhan (0.42-0.53, $p<0.05$) than in YRD (0.35, $p<0.05$), and lowest in BTH (0.27, $p<0.05$). The correlations indicate some capability for OMI to predict ozone daily variability on a statistical basis. The reduced major axis (RMA) regression slopes are consistent across the five regions and average 0.15 DU/ppb. We define an ozone episode day by afternoon concentrations exceeding 82 ppb, corresponding to the Chinese air quality standard. The bottom panel of Figure 2 compares the OMI ozone enhancements between episode and non-episode days as measured by the surface network. OMI is significantly higher ($p<0.05$) on episode days for all five regions.

## 4 OMI boundary layer sensitivity inferred from ozonesondes

The correlation of OMI with the MEE surface ozone data likely does not reflect a direct sensitivity of OMI to surface ozone, which is very weak, but rather a sensitivity to boundary layer ozone extending up to a certain depth and correlated with surface ozone. We examined in more detail the sensitivity of OMI to boundary layer ozone and its day-to-day variability by comparing to summertime 2005-2016 Hong Kong ozonesonde data. Figure 3a shows the measured ozonesonde profiles (in ppb) mapped on a 100 hPa grid and selecting only the days when concurrent OMI retrievals are available ($n = 57$). The boundary layer ozone (950-850 hPa) in the ozonesonde data has large day-to-day variability, ranging from 20 to 120 ppb with a mean of 47 ppb. The variability in the free troposphere is much less.

Figure 3b shows the ozonesonde data smoothed by the OMI averaging kernel sensitivities for the corresponding retrievals. The retrievals over Hong Kong have a mean DOFS of 0.46 below 400 hPa. We see from Figure 3c that the OMI information is weighted toward the free troposphere but there is sensitivity in the boundary layer, and since boundary layer variability is much larger it can make a major contribution to OMI variability. The L23 ozone smoothed from the ozonesonde data in Figure 3b has a correlation coefficient of 0.75 with the 950-850 hPa ozone in the original data. The temporal correlation coefficients of the OMI retrievals at different levels with the 950-850 hPa ozonesonde data are given in Figure 3d. The correlation coefficient with L23 OMI ozone is 0.51 ($p<0.05$), and the correlation coefficient with the 850-400 hPa OMI

ozone constructed by weighting the L24-L23-L22 retrievals is 0.50. Figure 3e shows a scatterplot of the latter. We see that high-ozone episodes in the 950-850 hPa sonde data are systematically associated with high OMI values, though the converse does not always hold because free tropospheric enhancements affecting OMI can also occur. For the 8 boundary layer episode days (> 82 ppb), the average OMI 850-400 hPa ozone is 23.7±3.1 DU, significantly higher than for the non-episode

5    days (18.2±4.1 DU). The Hong Kong ozonesonde data thus indicate that OMI can quantify the frequency of high-ozone episodes in the boundary layer even if it may not be reliable for individual events.

We applied the same daily correlation analysis to the other ozonesonde datasets and IAGOS aircraft measurements during 2005-2017 summers. For the 54 IAGOS vertical profiles coincident with OMI observations, the correlation coefficient of the

950 hPa in situ ozone and 850-400 hPa OMI ozone is $R = 0.59$ ($p<0.05$) (Figure S2). For the five ozonesonde sites with long-term observations, the correlation coefficients are 0.4-0.6 for Hanoi, Hong Kong, and Naha (south of 30°N), and 0-0.3 for Sapporo and Tsukuba (north of 35°N) (Figure S3), consistent with the patterns of daily correlations for the MEE data (Figure 1f).

The correlation between boundary layer ozone pollution and the OMI ozone retrievals could be due in part to correlation

between boundary layer and mid-tropospheric ozone, considering that both tend to be driven by the same weather systems. We used the ozonesonde data to examine what correlation with boundary layer (950-850 hPa) ozone would be observed if OMI were sensitive only to the free troposphere at ~500 hPa (where its sensitivity is maximum, Figure 3c) and not to the boundary layer. In that case the correlation coefficient $R_{1,3}$ of boundary layer ozone and the OMI 850-400 hPa retrievals would be given by (Vos, 2009):

$$R_{1,3} = R_{1,2}R_{2,3} \pm \sqrt{(1-R_{1,2}^2)(1-R_{2,3}^2)} \tag{2}$$

where $R_{1,2}$ is the correlation coefficient between boundary layer and 500 hPa ozone in the ozonesonde data, and $R_{2,3}$ is that between 500 hPa ozone and the OMI 850-400 hPa retrievals. As seen from Figure S4, $R_{1,3}$ at the five sonde sites is only ~0.2, implying that direct sensitivity to the boundary layer dominates the correlation of OMI with surface ozone at least in southern China. Further evidence for this is the ability of OMI to detect the ozone enhancements in megacity clusters (Figure

1).

We find that the low correlation of OMI with boundary layer ozone in the northern ozonesonde data is due not only to the low DOFS but also to a large variability of ozone in the upper troposphere. Figure 4 (left panel) shows the standard deviation of daily OMI 400-200 hPa ozone during 2005-2017 summers, indicating that upper tropospheric ozone has much higher variability in the north (> 34°N) than in the south. This is related to the location of the jet stream and more active

stratospheric influence (Hayashida et al., 2015). Figure 4 (right panel) displays the vertical profiles of ozone standard deviations for the five ozonesonde sites. For the two sites north of 34°N, the ozone variability becomes very large above 8

km. Since the OMI 850-400 hPa retrieval also contains information from above 400 hPa, this upper tropospheric variability causes a large amount of noise that masks the signal from boundary layer variability. For the three sites south of 34°N, the ozone variability in the boundary layer is much higher than in the free troposphere and the upper tropospheric ozone variability still remains low even above 8 km. In the rest of this paper we focus our attention on ozone episodes and the long-term trends in southern China (south of 34°N).

## 5. Using extreme value theory to predict the occurrence of high-ozone episodes from OMI data

We construct a point process (PP) model from extreme value theory (Cole, 2001) to estimate the likelihood of surface ozone exceeding a high-ozone threshold $u$ (here $u = 82$ ppb at 12-15 LT) at a given site $i$ and day $t$ given the observed OMI ozone enhancement $x_{i,t}$ for that day. The model describes the high tail of the ozone probability density function (pdf) as a Poisson process limit, conditioned on the local OMI observation. Such a model has been used previously to relate the probability of extreme air pollution conditions to meteorological predictor variables (Rieder et al., 2013; Shen et al., 2016, 2017; Pendergrass et al., 2019) but here we use the OMI enhancement as predictor variable. We fit the model to all daily concurrent observations of surface ozone and OMI ozone enhancements for the ensemble of eastern China sites south of 34°N in Figure 5 (90,601 observations for summers 2013-2017). The probability of exceeding the threshold at a site $i$ should depend not only on $x_{i,t}$ but also on its time-averaged value $\overline{x}_i$, because a high value of $\overline{x}_i$ means that a higher $x_{i,t}$ is less anomalous and more likely to represent an actual ozone exceedance than for a site with low $\overline{x}_i$. Thus the model has two predictor variables, $x_{i,t}$ and $\overline{x}_i$.

Details of the PP model can be found in Cole (2001). The model fits three parameters that control the shift, spread and shape of the high-tail pdf. The fit minimizes a cost function $L$ given by

$$L = \prod_{i=1}^{m} L_i(\mu_i, \sigma_i, \xi) \tag{3}$$

with

$$L_i(\mu_i, \sigma_i, \xi) = exp\{-\frac{1}{n_a}\sum_{t=1}^{n}[1+\frac{\xi(u-\mu_i)}{\sigma_{i,t}}]^{-1/\xi}\}\prod_{t=1}^{n}\{\frac{1}{\sigma_{i,t}}[1+\frac{\xi(y_{i,t}-\mu_i)}{\sigma_{i,t}}]^{-1/\xi-1}\}^{I[y_{i,t}>u]} \tag{4}$$

$$\mu_i = \alpha_0 + \alpha_1\overline{x}_i \tag{5}$$

$$\overline{x}_i = \frac{1}{n}\sum_{t=1}^{n}x_{i,t} \tag{6}$$

$$\sigma_{i,t} = \exp(\beta_0 + \beta_1 x_{i,t}) \tag{7}$$

Here $L_i(\mu_i, \sigma_i, \xi)$ is the cost function for site $i$ and $L$ is for the total cost function for all $m$ sites, $y_{i,t}$ is the daily 12-15 LT MEE surface ozone from each individual site $i$ on day $t$, $n_a = 92$ is the number of days in summer, $\mu_i$ is the location parameter for site $i$ conditioned on the 2013-2017 summertime mean OMI enhancements $\overline{x}_i$, $\sigma_{i,t}$ is the scale parameter conditioned on the local OMI ozone enhancements $x_{i,t}$, $\xi$ is the shape factor, and $I[y_{i,t} > u]$ is one if observed ozone is above the threshold and zero otherwise. Minimization of the cost function optimizes the values of the parameters $\alpha_0$, $\alpha_1$, $\beta_0$, $\beta_1$, and $\xi$ given the 90,601 $(x_{i,t}, y_{i,t})$ data pairs. The resulting values are $\alpha_0 = 103$ ppb, $\alpha_1 = 6.0$, $\beta_0 = 2.8$ ppb, $\beta_1 = -0.033$, $\xi = -0.12$. The probability of daily ozone exceeding the threshold $u$ is then calculated as

$$p(y_{i,t} \geq u \mid x_{i,t}) = \frac{1}{n_a}[1 + \xi(\frac{u - \mu_i}{\sigma_{i,t}})]^{-1/\xi} \qquad (8)$$

The model is optimized using the *extRemes* package in R (Gilleland and Katz, 2011). We performed a 10-fold cross validation of the model, in which we partitioned the sites into 10 equal subsets and repeatedly used one subset as testing data and the rest as training data. The results show that the predicted fraction of ozone episodes resembles that observed, with a spatial correlation of 0.62 (Figure 5a). The model tends to underestimate the probability of episodes in polluted regions due to the noise of daily OMI ozone. 82 ppb corresponds to the 84[th] percentile of the data, which is a relatively low threshold for application of extreme value theory. However, we find that the model can also accurately estimate the probability of exceedance above higher thresholds (Figure 5b) for the ensemble of eastern China sites south of 34°N, which confirms the property of threshold invariance of an extreme value model (Cole, 2001). We also tested the model with uniform location or scale factors, but neither could reproduce the observed spatial distribution of ozone episodes.

## 6. 2005-2017 trends in surface ozone inferred from OMI data

We used the long-term OMI ozone record for 2005-2017 to infer trends in surface ozone over southern China, not including any tropospheric background trends (removed by our subtraction of the North Pacific). Figure 6 shows the changes between 2005-2009 and 2013-2017 (an 8-year period) in mean summer afternoon ozone concentrations and in the number of high-ozone episode days per summer. Here we have extended the trend analysis to Taiwan because of the opportunity to compare to surface records. The changes in mean summer afternoon ozone concentrations are obtained from the difference in the mean OMI ozone enhancements between the two time periods (Figure S5) and applying equation (1). The changes in the number of high-ozone episode days per summer are obtained by applying the probability of exceeding 82 ppb (equation 8) to each pair of 5 years of OMI data. When averaged across southern China (including urban and rural regions), the mean summer afternoon ozone concentrations have increased by 3.5±3.0 ppb between the two periods (Figure 6a) and the number of ozone episodes (> 82 ppb) has increased by 2.2±0.4 days per summer (Figure 6b). Conditions have become particularly

worse in YRD and in Hubei, Guangxi, and Hainan provinces where the number of high-ozone days per summer has increased by more than 5.

We compared the OMI trends in Figure 6 to the trends of MDA8 ozone and number of high-ozone days reported by the long-term TOAR sites (Schultz et al., 2017) and our own analysis for the Hok Tsui station in Hong Kong (Wang et al., 2009). For Lin'an, Hong Kong, and the 5 sites in Taiwan (we report the mean value here), the changes of mean ozone concentrations from 2005-2009 to 2013-2017 are 1.1±3.6, 2.3±3.3, and -0.18±2.9 ppbv as estimated from OMI, compared to 0.7±3.6, 5.6±3.9 (or 5.8±1.3 in Hok Tsui station), and -0.75±2.5 ppbv for MDA8 ozone at the TOAR sites. The changes in the number of ozone episodes per summer are 1.2±0.7, 1.9±0.24, and -0.17±0.14 days in OMI, compared to 2.1±4.4, 1.8±1.7 (or 2.1±1.1 in Hok Tsui station), and -3.5±1.8 days at the TOAR sites. The standard errors are obtained by applying a parametric bootstrap method. The OMI inferred trends are generally consistent with the long-term records available from surface sites.

## 5 Discussion and conclusions

Satellite observations of tropospheric ozone in the UV could provide an indicator of surface ozone pollution if the associated boundary layer enhancement is large enough.  We presented a quantitative evaluation of this capability for OMI ozone retrievals in China by comparison to the extensive 2013-2017 ozone network data from the China Ministry of Ecology and Environment (MEE), together with vertical profiles from ozonesondes and aircraft. We went on to use the long-term OMI record (2005-2017) to infer surface ozone pollution trends over that period.

After subtracting the contribution from the North Pacific background, we find that the OMI enhancement over eastern China can reproduce the observed spatial distribution of multi-year mean summer afternoon ozone concentrations at the MEE sites, with a correlation coefficient $R = 0.73$ and a precision of 8 ppb. Even though OMI has little sensitivity to surface ozone, the high ozone levels seen at surface sites propagate deep enough in the boundary layer to be observed by OMI. Day-to-day correlation at individual sites is weaker ($R = 0.1$-$0.3$ north of $34^{o}$N, $0.3$-$0.6$ south of $34^{o}$N)) because of noise in individual OMI retrievals. But we find that OMI is statistically enhanced in urban areas when surface ozone exceeds an 8-h maximum daily average (MDA8) value of 82 ppb (the Chinese air quality standard).

To better understand the correlation of OMI with surface ozone we examined vertical ozone profiles from Hong Kong and other ozonesondes, and from the IAGOS commercial aircraft program.  Some of the correlation is driven by similar meteorology influencing ozone in the mid-troposphere (where OMI sensitivity is maximum) and the boundary layer, but most of the correlation is driven by direct sensitivity to the boundary layer. The Hong Kong ozonesonde data also indicates that OMI can quantify the frequency of high-ozone episodes in the boundary layer even if it may not be reliable for individual events. In southern China (< $34^{o}$N), we find that ozone variability in the tropospheric column is dominated by the

boundary layer, explaining the stronger correlations there. The lower correlation of OMI with surface ozone further north is due to large upper tropospheric variability in addition to lower sensitivity.

We went on to use the 2005-2017 OMI record to diagnose long-term trends of surface ozone in southern China (<34 $^{\circ}$N). This involved the development of a point process model from extreme value theory to infer the probability of surface ozone
exceeding 82 ppb and higher thresholds on the basis of the daily observed OMI ozone enhancements. The OMI record shows a general increase across southern China (including urban and rural regions) from 2005-2009 to 2013-2017 (8-year period) in mean summertime afternoon ozone (+3.5±3.0 ppb) and in the frequency of high-ozone episodes (+2.2±0.4 days per summer). Increases are particularly large in the Yangtze River Delta and in Hubei, Guangxi, and Hainan provinces. The trends are generally consistent with the few long-term records available from surface sites.

Our method for inferring ozone pollution and the frequency of high-ozone episodes from the OMI satellite data may be applied to other regions of the world where surface ozone is expected to be high but where in situ observations are lacking. The next generation of UV satellite instruments may improve this capability. The TROPOMI instrument launched in October 2017 is now providing daily observations with 3.5×7 km$^2$ pixel resolution, much finer than OMI (Theys et al., 2017). The GEMS (Geostationary Environment Monitoring Spectrometer) instrument is expected to launch in late 2019 and will
observe East Asia with hourly frequency and sensitivity similar to OMI (Bak et al., 2013). The TEMPO (Tropospheric Emissions: Monitoring of Pollution) geostationary satellite instrument to be launched around 2020 over North America will have a spectral range extending to the visible Chappuis bands where ozone detection sensitivity remains high down to the surface (Zoogman et al., 2011, 2017). This should allow for improved observations of surface ozone, particularly where concentrations are not as high as they are presently in China.

**Data availability**. The OMI ozone data and surface measurements are available upon request.

**Author contribution.** L. Shen and D. Jacob designed the experiments and L. Shen carried them out. X. Liu and G. Huang provided the satellite data. K. Li, H. Liao and T. Wang provided the surface observations. L. Shen and D. Jacob prepared the manuscript with contributions from all co-authors.

**Competing Interests**. The authors declare that they have no conflict of interest.

**Acknowledgments.** This work was funded by the Harvard Global Institute (HGI), by the NASA Earth Science Division, and by the Joint Laboratory for Air Quality and Climate (JLAQC) between Harvard and the Nanjing University for Information Sciences and Technology (NUIST).

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

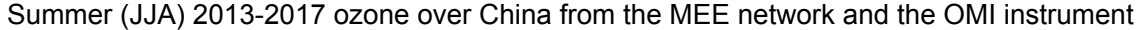

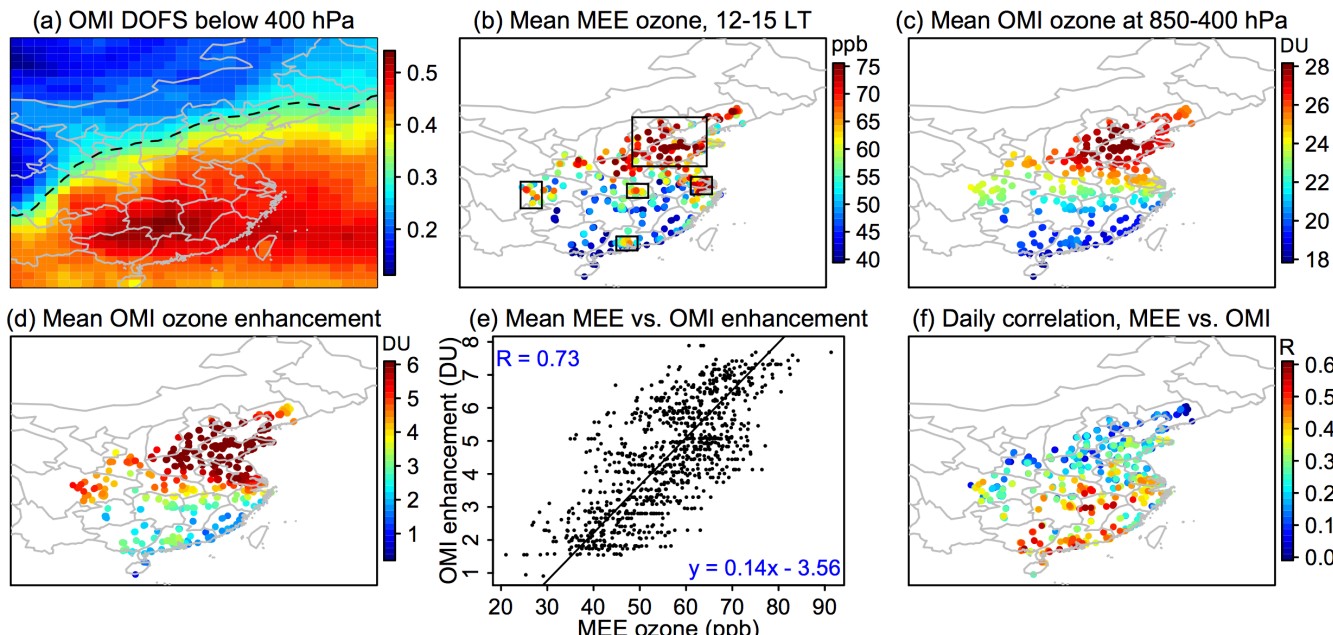

**Figure 1**. Summertime observations of ozone over China (JJA 2013-2017) from the MEE surface network and the OMI satellite instrument. (a) Mean degrees of freedom for signal (DOFS) of OMI ozone retrievals below 400 hPa. We limit our attention to the China domain with DOFS > 0.3 (south of dashed line) and to sites with at least 100 concurrent surface and OMI observations for the 2013-2017 period. (b) Mean midday (12-15 local time) ozone concentrations from the MEE surface network. Rectangles identify high-ozone regions discussed in the text including Beijing-Tianjing-Hebei (BTH, 114°-121°E, 34-41°E), Yangtze River Delta (YRD, 119.5°-121.5°E, 30-32.5°E), Pearl River Delta (PRD, 112.5°-114.5°E, 22-24°E), Sichuan Basin (SCB, 103.5°-105.5°E, 28-31.5°E), and Wuhan (113.5°-115.5°E, 29.5-31.5°E). (c) Mean OMI partial columns at 850-400 hPa. (d) Mean OMI ozone enhancements at 850-400 hPa after subtraction of the latitude-dependent mean background over the Pacific (150°E-150°W). (e) Spatial correlation of mean JJA 2013-2017 MEE ozone with the OMI ozone enhancement at 850-400 hPa. The correlation coefficient and the fitted reduced major axis (RMA) regression equation are shown inset. (f) Temporal correlation coefficients (*R*) of daily MEE surface ozone with OMI at individual sites, measuring the ability of OMI to capture the day-to-day variability of surface ozone.

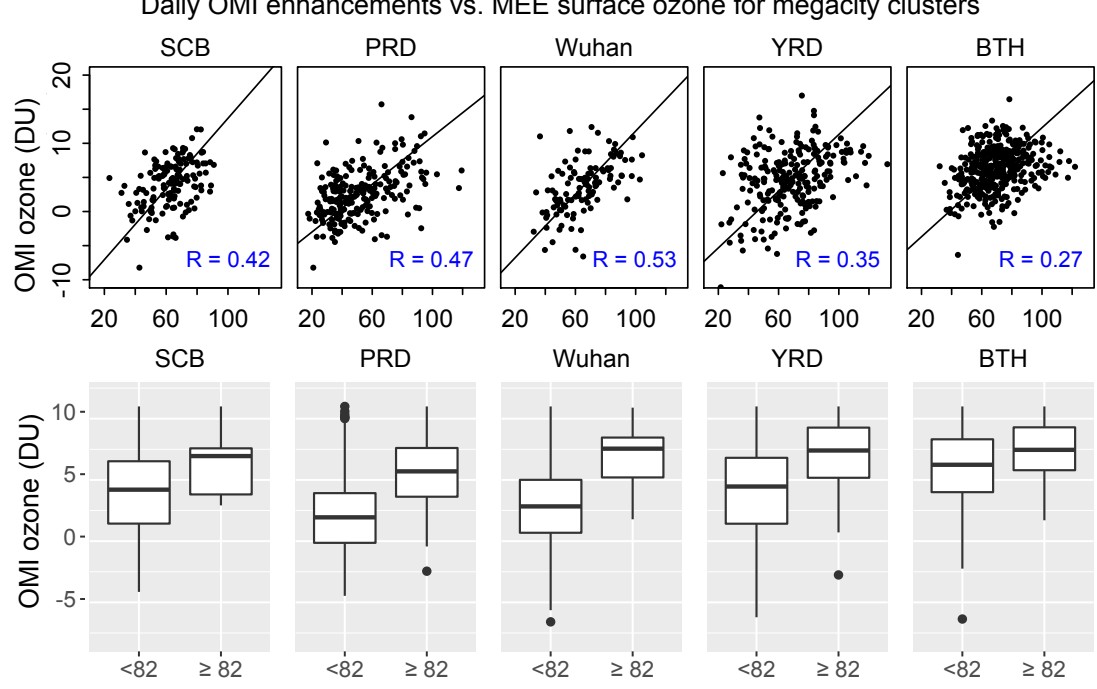

**Figure** 2. Ability of daily OMI observations to detect high-ozone episodes in the five megacity clusters of Figure 1. Daily surface afternoon (12-15 local time) observations from the MEE network in summer (JJA) 2013-2017 averaged over the megacity clusters are compared to the corresponding OMI enhancements relative to the Pacific background. The top panels show the correlations in the daily data, with correlation coefficients inset. Reduced-major-axis (RMA) linear regression lines are also shown. The bottom panels show the distributions of OMI enhancements for episode (≥ 82 ppbv) and non-episode (< 82 ppbv) days. The top and bottom of each box are the 25th and 75th percentiles, the centerline is the median, the vertical bars are the 2[th] and 98[th] percentiles, and the dots are outliers.

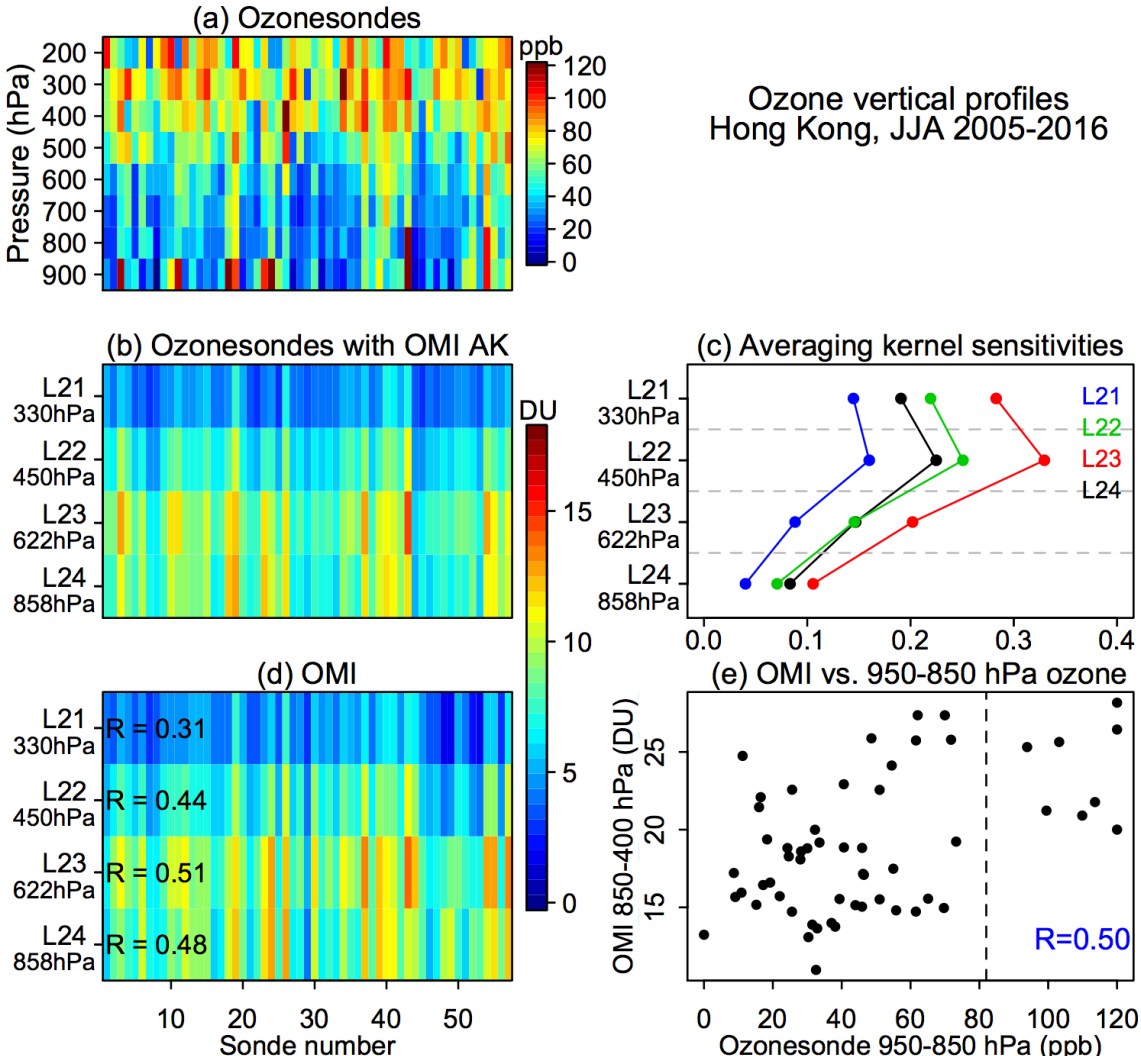

**Figure 3**. Ozone vertical profiles over Hong Kong in summer (JJA) 2015-2016. (a) Ozonesonde data coincident with OMI observations (*n*=57), averaged over a 100 hPa grid and arranged in chronological order. (b) The same ozonesonde data but smoothed by the OMI averaging kernels. Mean pressures for each OMI retrieval level are indicated. (c) Mean averaging kernel sensitivities for each OMI retrieval level, as described by the rows of the averaging kernel matrix; values are shown for August 2015 but are similar in other summer months and years. The dashed lines are boundaries between retrieval levels. (d) OMI ozone observations coincident with the ozonesondes. The correlations of unsmoothed 950-850 hPa ozonesonde data with the OMI retrievals for different levels are shown inset. (e) Relationship of unsmoothed 950-850 hPa ozonesonde data and OMI 850-400 hPa ozone. The correlation is shown inset. The dashed line corresponds to the Chinese ozone air quality standard (82 ppb).

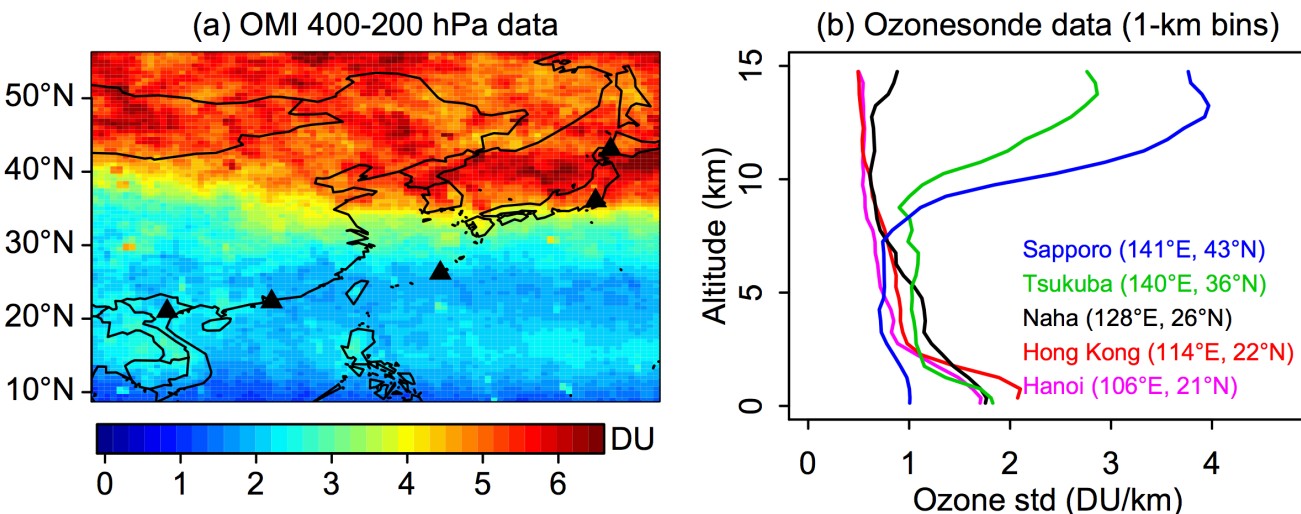

**Figure 4.** (a) Standard deviation of daily OMI 400-200 hPa ozone in East Asia during 2005-2017 summers. The triangles are the locations of ozonesonde sites with observations during this period. (b) Vertical profiles of daily ozone standard deviation in 1-km bins (DU/km) in the ozonesonde data for the 2005-2017 summers.

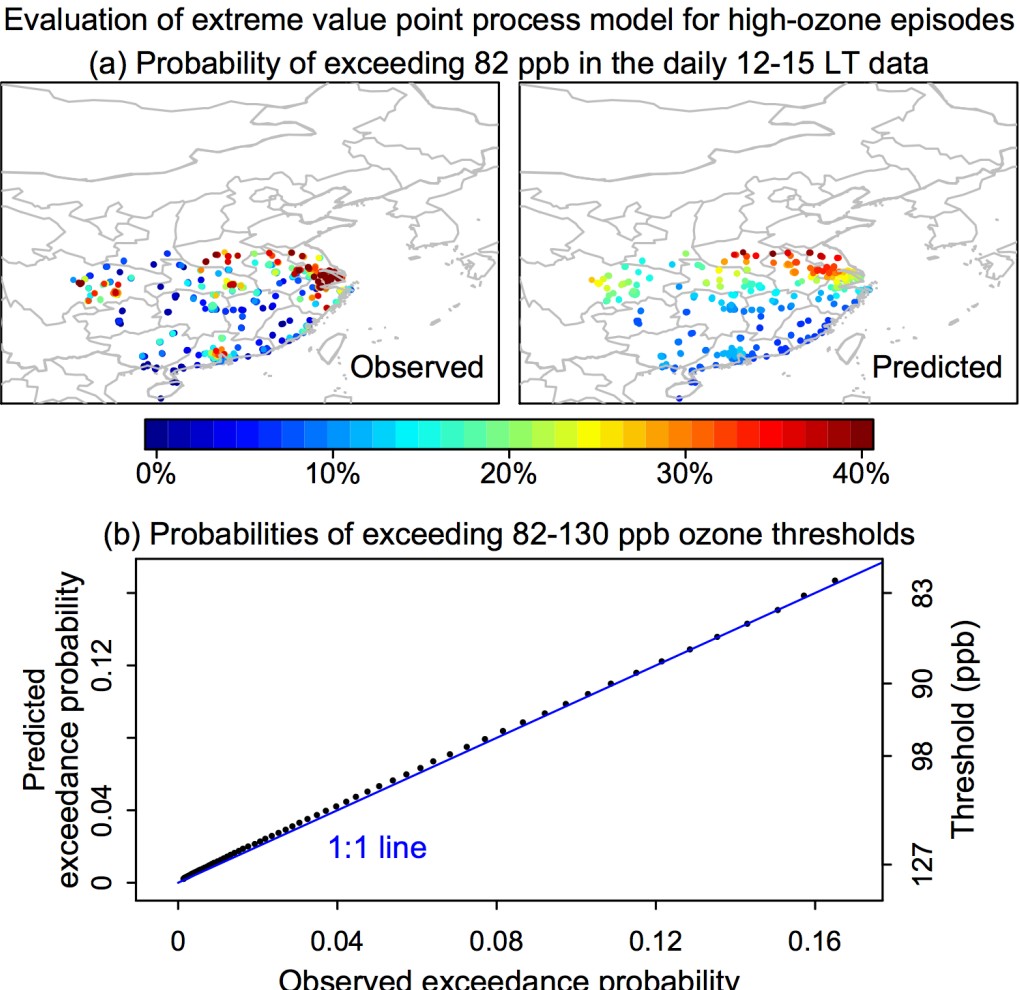

**Figure 5**. Evaluation of the extreme value point process (PP) model for predicting the probability of occurrence of summertime high-ozone episodes from the OMI daily data. The episodes are defined by exceedance of a given ozone threshold in the 3-hour average data at 12-15 local time. (a) Observed and predicted probability of ozone episode days exceeding a 82 ppb threshold. The predicted probability is calculated from equation (8). (b) Observed and predicted probabilities of exceeding higher thresholds from 82 to 130 ppb.

## Changes in summertime surface ozone pollution inferred from OMI
## (2005-2009 to 2013-2017)

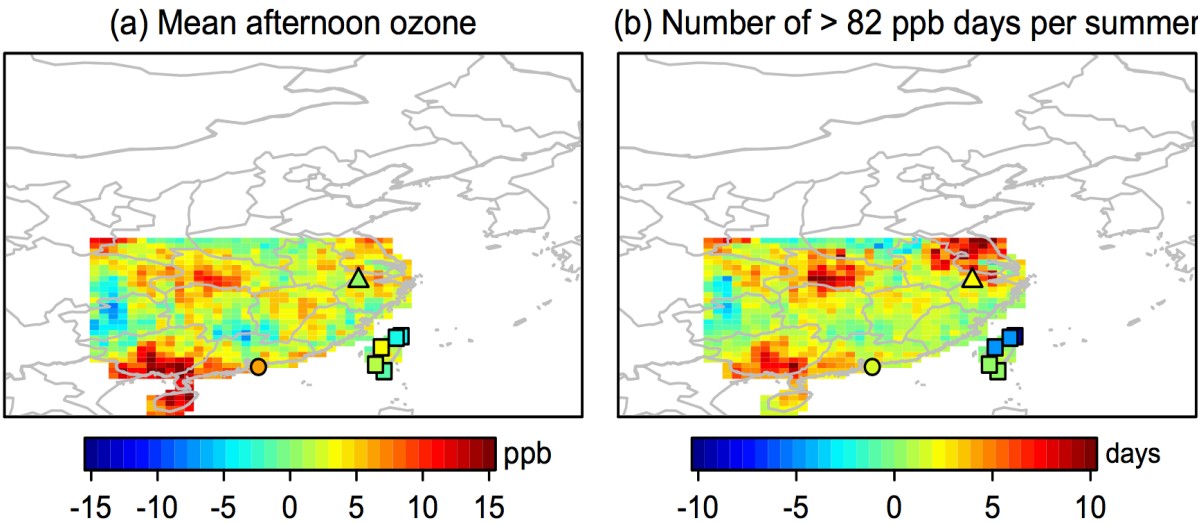

**Figure** 6. Changes in surface ozone pollution between the 2005-2009 and 2013-2017 periods (separated by 8 years) as inferred from OMI afternoon observations at around 13:30 local time. (a) Change in mean summer afternoon concentrations, obtained from the difference in the mean OMI ozone enhancements and applying equation (1). Also shown with symbols are observed changes in mean MDA8 ozone from in situ observations in Lin'an, Hong Kong, and Taiwan reported by TOAR (Schultz et al., 2017). Because the TOAR observations are only reported for 2005-2014, we estimate the changes from 2005-2009 to 2013-2017 on the basis of the reported linear trends during 2005-2014 (ppb a$^{-1}$). (b) Change in the number of high-ozone days (> 82 ppb) per summer, calculated by applying the probability of exceeding 82 ppb (equation 8) to the daily OMI enhancements. Also shown with symbols are observed changes of the number of days with MDA8 ozone exceeding 80 ppb at the TOAR sites, similarly adjusted as the change from 2005-2009 to 2013-2017.