# Peer review of "Figure S1 The sensitivity of retrieved ozone at L22-L24 to the ozone change at L24, as denoted by the columns of AK (Hayashida et al., 2015)."

_Atmospheric Chemistry and Physics, 2018_

## Referee Comment (RC1) · Anonymous Referee #1 · 6 Jan 2019

This paper seeks to quantify surface ozone across China using the SAO OMI tropospheric column ozone product. While I appreciate this effort to quantify such a relationship, the current analysis has not demonstrated a clear and convincing link between the lower/mid-tropospheric OMI retrievals and day-to-day ozone variability at the surface. I realize that the authors are trying to find some signal in OMI that reflects ozone at the surface, but the degrees of freedom are so small, and the sensitivity to surface ozone is so weak, that there's no real way to distinguish between the signal that comes from the surface and that which comes from 800 or 700 hPa. As presented, the relationship is more likely due to weather pattern variability causing ozone at the surface and in the free troposphere to vary in tandem. Far more work is required, including a

thorough evaluation of the OMI product against extensive IAGOS aircraft observations across mainland China, South Korea, Taiwan and Hong Kong. The additional analysis required to convince me that OMI can provide a meaningful evaluation of surface ozone across China goes beyond a standard major revision. My recommendation to the editor is that the paper be rejected to allow the authors adequate time to conduct additional product evaluation. If the expanded analysis can indeed demonstrate sensitivity of OMI to surface ozone then the authors will have the basis for a new manuscript which will make a valuable contribution to ozone monitoring across East Asia.

Further comments:

When I read the title and abstract I was under the impression that the authors had made a new breakthrough regarding the detection of surface ozone using OMI. It seemed like the instrument could actually detect ozone at the surface and the detection was so good that daily ozone variability at any given surface site could be determined with a precision of +/- 10.7 ppb. But when I read the full paper I learned that this is not the case.

The premise that lower/mid-tropospheric OMI ozone retrievals are closely associated with surface ozone is not shown in a convincing manner. The initial correlation of: [O3] = 8.9 $\Delta\Omega$ + 15.8 $\pm$ 10.7 appears to be driven entirely by the latitudinal gradient of ozone at the surface and in the mid-troposphere. Just because surface and OMI ozone have similar latitudinal gradients, when averaged over several years, does not mean that the mid- to lower troposphere can tell us how ozone varies at the surface from day to day. A better test of the relationship is to focus on a narrow latitude range. This is done in Figure 3, for daily observations, where we can see that the correlation is very low above five urban regions. For the region of Beijing the correlation is only R=0.27 which corresponds to an r-squared value of 0.07, which means that the variation in OMI only explains 7% of the ozone variability at the surface. The best case is made by the Wuhan region, but even here R=0.53, which means OMI only explains 28% of the surface ozone variability. Figure 3 shows that OMI is only weakly correlated with

surface ozone and provides no convincing argument that the retrieval is sensitive to surface ozone. The weak correlation is probably just due to weather patterns causing surface and lower to mid- tropospheric ozone to vary in tandem.

In a related comment, do the authors think that any correlation between ozone at the surface and ozone in the lower/mid troposphere is linked because of similar photochemical processes, or is the correlation just a coincidence due to meteorology? For example we know that in southern China the ozone at the surface varies strongly with the strength of the summertime Asian monsoon. When transport is from the south then the relatively clean air masses from the tropical Pacific bring air that is low in ozone, both at the surface and in the lower-mid troposphere. But when the monsoon winds weaken, mid-latitude air is allowed to move back into the region of southern China, bringing higher ozone to the lower and mid-troposphere. At the same time, the flow of clean air from the south also ceases at the surface, allowing ozone to build up in the polluted air masses from mainland China. Under this scenario ozone in the mid-troposphere is correlated with ozone at the surface even if the two layers are isolated from each other by strong temperature inversions.

OMI ozone could be compared to long-term ozone monitoring sites in rural areas which would be a better comparison than the urban data from the new Chinese monitoring network. It would be very helpful to see time series of daily OMI values (when available) and corresponding surface observations from the following sites: Mt Tai – data can be obtained from Prof. Likun Xue, Shandong University [Sun et al., 2016] Hok Tsui – located on the south coast of Hong Kong, data can be obtained from Prof. Tao Wang, Hong Kong Polytechnic [Wang et al., 2017] Shangdianzi – see Ma et al., 2016 LongFengShan – located in northeastern China. Contact Dr. Xiaobin Xu at the China Meteorological Administration: xiaobin_xu@189.cn LinAn – Near Shanghai, Contact Dr. Xiaobin Xu at the China Meteorological Administration: xiaobin_xu@189.cn XiangGeLiLa – in south central China, Contact Dr. Xiaobin Xu at the China Meteorological Administration: xiaobin_xu@189.cn

Ozone at the surface and in the mid-troposphere varies greatly with transport pathway and abrupt changes in air masses, and recent studies have shown that ozone in China varies with meteorology [Pu et al., 2017; Zhao et al., 2018]. The authors are aware of this phenomenon as their previous work has explored the impact of climate variability on ozone. Therefore I'm surprised that the authors didn't first explore how surface ozone across China varies with meteorology, such as surface temperatures (or temperature at 850 hpa) [Pusede et al., 2015], or with the height of the 500 hPa surface [Reddy et al., 2016], both of which correlate quite well with surface ozone. The authors should first determine the correlation between surface ozone and meteorology, and then compare these results to what they find from OMI ozone. Does OMI give more information on surface ozone than basic meteorology? Given that reanalysis data are available for all of China under all weather conditions (no cloud screening) I would think that the meteorology would perform better than OMI. If OMI performs less well than meteorology, is there any reason to use OMI to try to predict surface ozone, when meteorological analyses are available everywhere and at all times?

Another necessary analysis is to see if in situ observations of ozone in the mid-troposphere are correlated with surface ozone. I realize that the authors did look at ozonesonde profiles above Hong Kong, but they are not very frequent and they don't tell us anything about ozone in other parts of China, especially in the highly polluted North China Plain. The IAGOS program has hundreds of profiles above East Asia since 1995. As shown by Ding et al. [2005] and by Gaudel et al. [2018] ozone in summertime in the boundary layer is much greater than ozone in the mid-troposphere. The difference is due to very strong ozone production in the boundary layer, versus distant source regions for ozone in the mid-troposphere. If the authors conducted a transport study for ozone in the mid-troposphere they would find that very little of the air in this layer comes from the surface of China. Probably 80-90% of the mid-tropospheric above China air has either been in the mid-troposphere for days, or it comes from the boundary layer far upwind of China. The authors can freely access hundreds of commercial aircraft profiles of ozone and carbon monoxide above mainland China, Hong

Kong, Taiwan and South Korea from the IAGOS database. They can then apply the OMI averaging kernel to the profiles and determine the relationship between IAGOS ozone in the mid- and lower troposphere to ozone at the surface. Does IAGOS ozone in the mid-troposphere correlate with ozone at the surface? Is the correlation any better than when surface ozone is correlated with meteorology? Then compare the IAGOS relationship to the OMI relationship. Does OMI perform any better than IAGOS?

Figure 5 shows surface ozone trends across China which were derived from the OMI ozone product. The strongest trends are in the far north of China and in the far south of China. Based on the summer OMI trends (2005-2015) reported by the Tropospheric Ozone Assessment Report in supplementary Figure S-24 of Gaudel et al. [2018], OMI has a strong trend across southern China but no trend across northern China. Therefore I don't understand how Figure 5 can show trends across northern China. It would be helpful to include a map that shows the OMI trends across China.

References:

Ding, AJ, et al. 2008. Tropospheric ozone climatology over Beijing: analysis of aircraft data from the MOZAIC program. Atmos. Chem. Phys. 8: 1–13. DOI: https://doi.org/10.5194/acp-8-1-2008

Gaudel, A, et al. 2018. Tropospheric Ozone Assessment Report: Present-day distribution and trends of tropospheric ozone relevant to climate and global atmospheric chemistry model evaluation. Elem Sci Anth, 6: 39. DOI: https://doi.org/10.1525/elementa.291

Reddy, P.J. and Pfister, G.G., 2016. Meteorological factors contributing to the interannual variability of midsummer surface ozone in Colorado, Utah, and other western US states. Journal of Geophysical Research: Atmospheres, 121(5), pp.2434-2456.

Pu, X., Wang, T.J., Huang, X., Melas, D., Zanis, P., Papanastasiou, D.K. and Poupkou, A., 2017. Enhanced surface ozone during the heat wave of 2013 in Yangtze River Delta

region, China. Science of the Total Environment, 603, pp.807-816.

Pusede, S.E., Steiner, A.L. and Cohen, R.C., 2015. Temperature and recent trends in the chemistry of continental surface ozone. Chemical reviews, 115(10), pp.3898-3918.

Sun, L., L. Xue, et al. (2016), Significant increase of summertime ozone at Mount Tai in Central Eastern China, Atmos. Chem. Phys., 16, 10637–10650.

Wang, T., et al. (2017), Ozone pollution in China: A review of concentrations, meteorological influences, chemical precursors, and effects, Science of the Total Environment, 575, 1582–1596.

Zhao, S., Yu, Y., Yin, D., Qin, D., He, J. and Dong, L., 2018. Spatial patterns and temporal variations of six criteria air pollutants during 2015 to 2017 in the city clusters of Sichuan Basin, China. Science of The Total Environment, 624, pp.540-557.

---

## Referee Comment (RC2) · Anonymous Referee #2 · 20 Jan 2019

This is a nice study that explores the potential of OMI observations of tropospheric ozone to detect the ozone pollution over China. While it's unrealistic to use OMI data to capture the day-to-day variability of ozone pollution, the authors show extreme ozone pollution may be detectable by aggregating long-term observations using statistical methods. Overall I think this is an important study to the field, which opens up the possibility to use satellite observations to detect surface ozone pollution, but I think the authors overpromise the value of satellite data. I have several major concerns:

1. My major concern is that the authors seem to overpromise the value of OMI data for characterizing the spatial and temporal trend of ground-level ozone. The title and the

abstract leave me an impression that OMI satellite data can capture the spatial distribution and the long-term trend in ground-level ozone, but the results only suggest OMI may be able to detect high ozone pollution and capture the large-scale or latitudinal variations. I suggest the authors consider revising the title, otherwise it'd be misleading to readers. The authors need to be more careful with the wording. I think this work would actually be much more valuable if the authors can clarify the limitations of OMI data, which will also be useful for preparation of next-generation satellites.

2. Is the point process model you used to predict ozone exceedance probability site specific? If so, how can you apply this method widely to areas without ground-based sites (as you promised in the conclusion)? The authors present the surface ozone pollution and exceedance probability only at ground-based sites, but why not show the distribution across China? For example, MEE network mainly consists of urban sites. Can you use OMI data to tell the spatial patterns of ozone pollution over rural/remote areas? If not, what's the added value of OMI data to existing ground-based network?

3. Figure 5: While OMI data may be able to detect the sign of the change in ground-level ozone, the magnitude of the change is less convincing to me. The authors suggest a 0.67 ppb /year increase in mean ozone over China, which seems to be lower than previous studies. The point process model is trained with ground-based observations in 2013-2017, but it's unknown how the model performs for early years 2005 - 2009. I'd suggest the authors use available long-term ground-based ozone observations to verify the long-term change. I understand long-term ground-based observations are not generally available over China, but since the OMI data are global, it's possible to extend the analysis to wider regions (e.g. Hong Kong, Japan) where long-term sites are available for evaluation.

---

## Referee Comment (RC3) · Anonymous Referee #3 · 22 Jan 2019

This paper explored the capability of OMI ozone columns to represent the surface O3. I feel the satellite data is over-interpreted based on the evidence provided in the paper. However, I do believe it will be big news if substantial improvements are made to prove that the conclusion is solid.

General comments: 1. The sensitivity of OMI O3 to the lower troposphere is very low. I feel that is the reason why no quantitative comparison to surface observations has so far been done. I'm wondering is there any improvements that have been made to make the quantitative comparison robust? Why does not the quantitative comparison work for other regions, but work for China? 2. The robustness of the residual. How large is

the temporal and spatial variations of the background? Is it likely that such variations bring significant uncertainties to the subtraction? 3. The correlation between MEE and OMI. The correlation seems to be related with the dependence of O3 on latitude. I suggest additional analysis here to prove that is not the case.

Specific comments: 1. "We exclude outliers with over 35 Dobson Units (DU) at 850-400 hPa (>99th percentile in eastern China) and exclude July 2011 when the retrievals are anomalously high." Please give the reference to the exclusion. Otherwise, please quantify the influence of the exclusion. 2. "We see that high-ozone episodes in the 950-850 hPa sonde data are systematically associated with high OMI values, though the converse does not always hold." Additional explanation for the reason is expected.

---

## Author Comment (AC2) · 19 Mar 2019

**Response to referee comments on "Spatial distribution and temporal trend of ozone pollution**
**in China observed with the OMI satellite instrument, 2005–2017"**
We thank the referees for their careful reading of the manuscript and the valuable comments. This
document is organized as follows: the Referee's comments are in *italic*, our responses are in plain
text, and all the revisions in the manuscript are shown in blue. The line numbers in this document
refer to the updated manuscript.

## Referee #2

*This is a nice study that explores the potential of OMI observations of tropospheric ozone to detect*
*the ozone pollution over China. While it's unrealistic to use OMI data to capture the day-to-day*
*variability of ozone pollution, the authors show extreme ozone pollution may be detectable by*
*aggregating long-term observations using statistical methods. Overall I think this is an important*
*study to the field, which opens up the possibility to use satellite observations to detect surface ozone*
*pollution, but I think the authors overpromise the value of satellite data. I have several major*
*concerns:*
**Response**. Thanks for raising these good points. This feedback has significantly improved the
manuscript. Now we have a new Figure 4 showing that OMI 850-400 retrievals have limited skill in
predicting the daily ozone variability in the north and we only predict the trends of ozone pollution in
southern China (south of 34°N). We have new in-situ observations to validate the trends inferred
from the OMI, which are shown in Figure 6.
1. *My major concern is that the authors seem to overpromise the value of OMI data for*
*characterizing the spatial and temporal trend of ground-level ozone. The title and the abstract leave*
*me an impression that OMI satellite data can capture the spatial distribution and the long-term trend*
*in ground-level ozone, but the results only suggest OMI may be able to detect high ozone pollution*
*and capture the large-scale or latitudinal variations. I suggest the authors consider revising the title,*
*otherwise it'd be misleading to readers. The authors need to be more careful with the wording. I*
*think this work would actually be much more valuable if the authors can clarify the limitations of*
*OMI data, which will also be useful for preparation of next-generation satellites.*
**Response**. Thanks for making such a good point. Now we revised the title and also discussed the
limitations in many parts of the main text.
New title. Ability of the OMI satellite instrument to observe surface ozone pollution in China:
application to 2005-2017 ozone trends
P1 L18. OMI is much more successful at capturing the day-to-day variability of surface ozone at sites in
southern China <34°N ($R$ = 0.3-0.6) than in northern China ($R$ = 0.1-0.3) because of weaker retrieval
sensitivity and larger upper tropospheric variability in the north.
P5 L7. This implies that OMI can only provide statistical rather than deterministic temporal information on
ozone pollution episodes, and may be more useful in South than in North China. We return to this point in
Section 4.
P5 L18. The correlation of OMI with the MEE surface ozone data likely does not reflect a direct sensitivity of

OMI to surface ozone, which is very weak, but rather a sensitivity to boundary layer ozone extending up to a
certain depth and correlated with surface ozone.
P6L27. We find that the low correlation of OMI with boundary layer ozone in the northern ozonesonde data is
due not only to the low DOFS but also to a large variability of ozone in the upper troposphere. Figure 4 (left
panel) shows the standard deviation of daily OMI 400-200 hPa ozone during 2005-2017 summers, indicating
that upper tropospheric ozone has much higher variability in the north (> 34°N) than in the south. This is
related to the location of the jet stream and more active stratospheric influence (Hayashida et al., 2015). Figure
4 (right panel) displays the vertical profiles of ozone standard deviations for the five ozonesonde sites. For the
two sites north of 34°N, the ozone variability becomes very large above 8 km. Since the OMI 850-400 hPa
retrieval also contains information from above 400 hPa, this upper tropospheric variability causes a large
amount of noise that masks the signal from boundary layer variability.  For the three sites south of 34°N, the
ozone variability in the boundary layer is much higher than in the free troposphere and the upper tropospheric
ozone variability still remains low even above 8 km. In the rest of this paper we focus our attention on ozone
episodes and the long-term trends in southern China (south of 34°N).
*2. Is the point process model you used to predict ozone exceedance probability site specific? If so,*
*how can you apply this method widely to areas without ground-based sites (as you promised in the*
*conclusion)? The authors present the surface ozone pollution and exceedance probability only at*
*ground-based sites, but why not show the distribution across China? For example, MEE network*
*mainly consists of urban sites. Can you use OMI data to tell the spatial patterns of ozone pollution*
*over rural/remote areas? If not, what's the added value of OMI data to existing ground-based*
*network?*
**Response**. Thanks. The point process model makes use of all the data. Now we show the trends of
ozone for all rural and remote regions in south China.
P7 L14. We fit the model to all daily concurrent observations of surface ozone and OMI ozone enhancements
for the ensemble of eastern China sites in Figure 1 (90,601 observations for summers 2013-2017).

[Figure]

**Changes in summertime surface ozone pollution inferred from OMI**
**(2005-2009 to 2013-2017)**

**Figure** 6. Changes in surface ozone pollution in China between 2005-2009 and 2013-2017 as
inferred from OMI afternoon observations at around 13:30 local time. (a) Change in mean summer
afternoon concentrations, obtained from the difference in the mean OMI enhancements at 850-400

hPa and applying equation (1). Also shown with symbols are observed changes in mean MDA8
ozone from in situ observations in Lin'an, Hong Kong, and Taiwan reported by TOAR  (Schultz et
al., 2018). Because the TOAR observations are only reported for 2005-2014, we estimate the
changes from 2005-2009 to 2013-2017 on the basis of the reported linear trends during 2005-2014
(ppb a$^{-1}$). The change of 12-15 LT ozone at the Hok Tsui station in Hong Kong is 5.8 ppb.  (b)
Change in the number of high-ozone days (> 82 ppb) per summer, calculated by applying the
probability of exceeding 82 ppb (equation 8) to the daily OMI enhancements. Also shown with
symbols are observed changes of the number of days with MDA8 ozone exceeding 80 ppb at the
TOAR sites, similarly adjusted as the change from 2005-2009 to 2013-2017. The change in the
number of days with 12-15 LT ozone exceeding 82 ppbv at the Hok Tsui station in Hong Kong is
2.1 days.

*3. Figure 5: While OMI data may be able to detect the sign of the change in ground- level ozone, the*
*magnitude of the change is less convincing to me. The authors suggest a 0.67 ppb /year increase in*
*mean ozone over China, which seems to be lower than previous studies. The point process model is*
*trained with ground-based observations in 2013-2017, but it's unknown how the model performs for*
*early years 2005 - 2009. I'd suggest the authors use available long-term ground-based ozone*
*observations to verify the long-term change.  I understand long-term ground-based observations are*
*not generally available over China, but since the OMI data are global, it's possible to extend the*
*analysis to wider regions (e.g. Hong Kong, Japan) where long-term sites are available for*
*evaluation.*
**Response**. Thanks. We have new in-situ observations from TOAR and also from a Hong Kong site to
validate the trends inferred from the OMI, which are shown in Figure 6. We find the OMI inferred
trends are fairly consistent with the long-term records available from surface sites. We also add discussion
in the main text.
P4 L11. For evaluating the long-term surface ozone trends inferred from OMI, we use 2005-2014 trend
statistics for maximum daily 8-hour average (MDA8) ozone from the Tropospheric Ozone Assessment Report
(TOAR) (Schultz et al., 2018). We also have 2005-2017 JJA 12-15 LT mean ozone at the Hok Tsui station in
Hong Kong (Wang et al., 2009).
P9 L5. We compared the OMI trends in Figure 6 to the trends of MDA8 ozone and number of high-ozone days
reported by the long-term TOAR sites (Schultz et al., 2018) and our own analysis for the Hok Tsui station in
Hong Kong (Wang et al., 2009). For Lin'an, Hong Kong, and the 5 sites in Taiwan, the changes of mean
ozone concentrations from 2005-2009 to 2013-2017 are 1.1, 2.3, and -0.18±2.2 ppbv (standard deviation
among the 5 sites) as estimated from OMI, compared to 0.7, 5.6 (or 5.8 in Hok Tsui station), and -0.75±3.4
ppbv for MDA8 ozone at the TOAR sites. The changes  in the number of ozone episodes per summer are 1.2,
1.9, and -0.17±0.74 days in OMI, compared to 2.1, 1.8 (or 2.1 in Hok Tsui station), and -3.5±3.9 days at the
TOAR sites. These OMI inferred trends are fairly consistent with the long-term records available from surface
sites.

---

## Author Response (AR1)

**Response to referee comments on "Spatial distribution and temporal trend of ozone pollution**
**in China observed with the OMI satellite instrument, 2005–2017"**
We thank the referees for their careful reading of the manuscript and the valuable comments. This
document is organized as follows: the Referee's comments are in *italic*, our responses are in plain
text, and all the revisions in the manuscript are shown in blue. The line numbers in this document
refer to the updated manuscript.

**Referee #1**

*This paper seeks to quantify surface ozone across China using the SAO OMI tropospheric column*
*ozone product. While I appreciate this effort to quantify such a relationship, the current analysis has*
*not demonstrated a clear and convincing link between the lower/mid-tropospheric OMI retrievals*
*and day-to-day ozone variability at the surface. (1) I realize that the authors are trying to find some*
*signal in OMI that reflects ozone at the surface, but the degrees of freedom are so small, and the*
*sensitivity to surface ozone is so weak, that there's no real way to distinguish between the signal that*
*comes from the surface and that which comes from 800 or 700 hPa. (2) As presented, the*
*relationship is more likely due to weather pattern variability causing ozone at the surface and in the*
*free troposphere to vary in tandem. (3) Far more work is required, including a thorough evaluation*
*of the OMI product against extensive IAGOS aircraft observations across mainland China, South*
*Korea, Taiwan and Hong Kong. The additional analysis required to convince me that OMI can*
*provide a meaningful evaluation of surface ozone across China goes beyond a standard major*
*revision. My recommendation to the editor is that the paper be rejected to allow the authors*
*adequate time to conduct additional product evaluation. If the expanded analysis can indeed*
*demonstrate sensitivity of OMI to surface ozone then the authors will have the basis for a new*
*manuscript which will make a valuable contribution to ozone monitoring across East Asia.*

**Response**. Thanks for the careful reading of our manuscript and raising so many good points. The
feedback has significantly improved our work. We also wish to draw the reviewer's attention to that
we have revised the title, discussed the limitations of this work and also validated the OMI inferred
ozone trends with surface observations (Figure 6).
New title. Ability of the OMI satellite instrument to observe surface ozone pollution in China:
application to 2005-2017 ozone trends.

Since this is a long comment, we decompose it into three parts and answer each part one by one.

(1) Yes, the reviewer made a very good point here that the OMI cannot distinguish the signal that
comes from the surface and from the lower troposphere (e.g. 800 and 700 hPa), given the relatively
low vertical resolution of the retrievals. We have made these changes in the text to reduce the
confusion.

P5 L18. The correlation of OMI with the MEE surface ozone data likely does not reflect a direct sensitivity of
OMI to surface ozone, which is very weak, but rather a sensitivity to boundary layer ozone extending up to a
certain depth and correlated with surface ozone.

The DOFS depends on what error is assumed in the prior estimate. Since the prior from McPeters et
al. (2007) has low boundary layer ozone with low associated error then the DOFS would
underestimate the ability to observe the polluted boundary layer. Now we say this.
P3L19. Even though a DOFS of 0.3 is still low, it is based on the prior estimate of low boundary layer ozone
in the McPeters et al. (2007) zonal mean climatology.

Also the ozone sonde observations show that the ozone variability in the boundary layer is 80-100%
higher than in the free troposphere. In more polluted regions like in mega city clusters, this difference
should be even larger. These results indicate that the boundary layer ozone is more likely to drive the
daily variability of OMI 850-400 hPa retrievals in regions like South China. We have added the
following figure.

[Figure]

**Figure 4.** (a) Standard deviation of daily OMI 400-200 hPa ozone in East Asia during 2005-2017
summers. The triangles are the locations of ozonesonde sites with observations during this period. (b)
Vertical profiles of daily ozone standard deviation in 1-km bins (DU/km) in the ozonesonde data for
the 2005-2017 summers.

P6 L26. We find that the low correlation of OMI with boundary layer ozone in the northern
ozonesonde data is due not only to the low DOFS but also to a large variability of ozone in the
upper troposphere. Figure 4 (left panel) shows the standard deviation of daily OMI 400-200 hPa
ozone during 2005-2017 summers, indicating that upper tropospheric ozone has much higher
variability in the north (> 34°N) than in the south. This is related to the location of the jet stream
and more active stratospheric influence (Hayashida et al., 2015). Figure 4 (right panel) displays the
vertical profiles of ozone standard deviations for the five ozonesonde sites. For the two sites north
of 34°N, the ozone variability becomes very large above 8 km. Since the OMI 850-400 hPa
retrieval also contains information from above 400 hPa, this upper tropospheric variability causes a
large amount of noise that masks the signal from boundary layer variability.  For the three sites
south of 34°N, the ozone variability in the boundary layer is much higher than in the free
troposphere and the upper tropospheric ozone variability still remains low even above 8 km. In the
rest of this paper we focus our attention on ozone episodes and the long-term trends in southern
China (south of 34°N).

(2) We thank the reviewer for pointing out this issue. But based on our analysis, the weather
patterns are insufficient to explain the observed relationship. We have conducted a sensitivity
experiment and made these changes in the text.

P 6 L15. The correlation between boundary layer ozone pollution and the OMI ozone retrievals
could be due in part to correlation between boundary layer and mid-tropospheric ozone, considering
that both tend to be driven by the same weather systems. We used the ozonesonde data to examine
what correlation with boundary layer (950-850 hPa) ozone would be observed if OMI were
sensitive only to the free troposphere at ~500 hPa (where its sensitivity is maximum, Figure 3c) and
not to the boundary layer. In that case the correlation coefficient $R_{1,3}$ of boundary layer ozone and
the OMI 850-400 hPa retrievals would be given by (Vos, 2009):

$$R_{1,3} = R_{1,2}R_{2,3} \pm \sqrt{(1 - R_{1,2}^2)(1 - R_{2,3}^2)} \qquad (2)$$

where $R_{1,2}$ is the correlation coefficient between boundary layer and 500 hPa ozone in the
ozonesonde data, and $R_{2,3}$ is that between 500 hPa ozone and the OMI 850-400 hPa retrievals. As
seen from Figure S4, $R_{1,3}$ at the five sonde sites is only ~0.2, implying that direct sensitivity to the
boundary layer dominates the correlation of OMI with surface ozone at least in southern China.
Further evidence for this is the ability of OMI to detect the ozone enhancements in megacity
clusters (Figure 1).

[Figure]

**Figure S4**. Correlation of OMI 850-400 hPa ozone and the boundary layer ozone assuming that the
OMI were sensitive only to the free troposphere at ~500 hPa (where its sensitivity is maximum,
Figure 3c) and not to the boundary layer. The black line is the correlation of ozone at different
pressure levels with 500 hPa ozone in the sonde observations. The blue line is the estimated
correlation of OMI 850-400 hPa with ozone at different layers if the satellite can only detect the
signal at 500 hPa but not from other layers, as calculated using Equation 2. See text for more details.

(3) We have processed all IAGOS aircraft observations in East Asia during 2005-2017 summers
and we only find 54 profiles in 8 airports that can be used to validate the OMI. Given so few
profiles, it is unlikely to evaluate the long-term correlation of surface and mid-tropospheric ozone
at each airport. But combining all profiles together, the temporal correlation coefficients of the 950
hPa ozone and 850-400 hPa OMI ozone is 0.59. These results are also consistent with what we find
in the Hong Kong sonde site (Figure 2). We have made these changes.

P6 L9. We applied the same daily correlation analysis to the other ozonesonde datasets and IAGOS
aircraft measurements during 2005-2017 summers. For the 54 IAGOS vertical profiles coincident
with OMI observations, the correlation coefficient of the 950 hPa in situ ozone and 850-400 hPa
OMI ozone is $R = 0.59$ ($p<0.05$) (Figure S2).

[Figure]

**Figure S2**. (a) Location and the number of tropospheric profiles at each airport in IAGOS that are
coincident with OMI retrievals. We only select these profiles between 12-15 local time. (b) Ozone
profiles from IAGOS, but mapped to OMI layers. The missing data at L21 is in part because we
only select pixels that are within 200 km from the airport on the flight path. (c) OMI ozone profiles
coincident with the ozonesondes. The correlations of unsmoothed 950 hPa ozone data in IAGOS
with the OMI retrievals for different levels are shown inset. The correlation with 850-400 hPa OMI
ozone is 0.59 ($p<0.05$)

*Further comments:*

*When I read the title and abstract I was under the impression that the authors had made a new*

*breakthrough regarding the detection of surface ozone using OMI. It seemed like the instrument*
*could actually detect ozone at the surface and the detection was so good that daily ozone variability*
*at any given surface site could be determined with a precision of +/- 10.7 ppb. But when I read the*
*full paper I learned that this is not the case.*

*The premise that lower/mid-tropospheric OMI ozone retrievals are closely associated with surface*
*ozone is not shown in a convincing manner. The initial correlation of: [O3]= 8.9 ΔΩ + 15.8 ± 10.7*
*appears to be driven entirely by the latitudinal gradient of ozone at the surface and in the mid-*
*troposphere. Just because surface and OMI ozone have similar latitudinal gradients, when averaged*
*over several years, does not mean that the mid- to lower troposphere can tell us how ozone varies at*
*the surface from day to day. A better test of the relationship is to focus on a narrow latitude range.*
*This is done in Figure 3, for daily observations, where we can see that the correlation is very low*
*above five urban regions. For the region of Beijing the correlation is only R=0.27 which corresponds*
*to an r-squared value of 0.07, which means that the variation in OMI only explains 7% of the ozone*
*variability at the surface. The best case is made by the Wuhan region, but even here R=0.53, which*
*means OMI only explains 28% of the surface ozone variability. Figure 3 shows that OMI is only*
*weakly correlated with surface ozone and provides no convincing argument that the retrieval is*
*sensitive to surface ozone. The weak correlation is probably just due to weather patterns causing*
*surface and lower to mid- tropospheric ozone to vary in tandem.*

**Response**. Yes, the reviewer has made a very good point here that the satellite has a lot of noise. This
is a common problem when we use satellite data.  But noisy data still contains information, and we
can reduce the noise by either temporally averaging the data over multiple years, or training the
model with a lot of data together. In our study, we tried both ways. When we average the data over a
five-year period, the OMI 850-400 hPa ozone displays a strong correlation (*R*=0.73) with the surface
observations. When we fit all the daily data in eastern China together, we find the extreme value
model can well simulate the distribution of high ozone concentrations. We also find the resulting
model can accurately estimate the probability of higher thresholds, which is a strong signal that our
extreme value model is well fitted.

The reviewer is correct that our old manuscript indeed overpromised the value of OMI data,
especially in the northern China. Due to the stronger jet wind activities and more active stratosphere-
troposphere exchange in the north, the upper tropospheric ozone there has much higher daily
variability, making OMI 850-400 hPa ozone less reliable in predicting the daily episode and inferring
the long-term trends. The strong jet wind also means these regions are very sensitive to global
background. At the same time, OMI has relatively low sensitivity in the north. So in our revised
manuscript, we don't fit the extreme value model or predict the trends in the north. We have added a
new figure 4 and more discussions at P6L26-P7L6. Based on reviewer's suggestion, we have
changed the title (see new title) and also discussed the limitations of this work (see many blue
highlighted text in the revised manuscript).

But the OMI data should still be useful in south China for these reasons. First, OMI has higher
sensitivity in the South, and the daily correlation of surface ozone and OMI 850-400 retrievals is statistically significant for most sites (Figure 1). Second, the upper tropospheric ozone variability is
much smaller and the boundary ozone variability can be a strong modulator of OMI 850-400 hPa
ozone variability (new figure 4). Third, the inferred long-term trends of surface ozone from OMI
fairly agree with these from TOAR sites (Figure 6).
We also make it clear we have removed the latitude-dependent background.
P3 L23. To remove this gradient and also any long-term uniform drift in the data, we subtract the monthly
mean Pacific background (150$^{o}$E-150$^{o}$W) for the corresponding latitude and month
P4 L22. After subtracting the North Pacific background for the corresponding latitude in month, we obtain the
OMI ozone enhancements shown in Figure 1d.
P4 L23. The spatial correlation coefficient between the OMI ozone enhancements and the MEE surface
network is $R = 0.73$ over eastern China. The correlation is driven in part by the latitudinal gradient but also by
the enhancements in the large megacity clusters identified as rectangles in Figure 1b. Thus the correlation
coefficient is $R = 0.55$ for the 26-34°N latitude band including YRD, SCB, and Wuhan.
*In a related comment, do the authors think that any correlation between ozone at the surface and*
*ozone in the lower/mid troposphere is linked because of similar photo- chemical processes, or is the*
*correlation just a coincidence due to meteorology? For example we know that in southern China the*
*ozone at the surface varies strongly with the strength of the summertime Asian monsoon. When*
*transport is from the south then the relatively clean air masses from the tropical Pacific bring air*
*that is low in ozone, both at the surface and in the lower-mid troposphere. But when the monsoon*
*winds weaken, mid-latitude air is allowed to move back into the region of southern China, bringing*
*higher ozone to the lower and mid-troposphere. At the same time, the flow of clean air from the south*
*also ceases at the surface, allowing ozone to build up in the polluted air masses from mainland*
*China. Under this scenario ozone in the mid- troposphere is correlated with ozone at the surface*
*even if the two layers are isolated from each other by strong temperature inversions.*
**Response**. Thanks for pointing out this issue. But meteorology is insufficient to explain the observed
correlation of surface ozone and OMI 850-400 hPa ozone. Please check L75-100 of this response
letter for more details (or P5 L15-25 in the main text).
*OMI ozone could be compared to long-term ozone monitoring sites in rural areas which would be a*
*better comparison than the urban data from the new Chinese monitoring network. It would be very*
*helpful to see time series of daily OMI values (when avail- able) and corresponding surface*
*observations from the following sites: Mt Tai – data can be obtained from Prof. Likun Xue, Shandong*
*University [Sun et al., 2016] Hok Tsui – located on the south coast of Hong Kong, data can be*
*obtained from Prof. Tao Wang, Hong Kong Polytechnic [Wang et al., 2017] Shangdianzi – see Ma et*
*al., 2016 LongFengShan – located in northeastern China. Contact Dr. Xiaobin Xu at the China*
*Meteorological Administration: xiaobin_xu@189.cn LinAn – Near Shanghai, Contact Dr. Xiaobin*
*Xu at the China Meteorological Administration: xiaobin_xu@189.cn Xi- angGeLiLa – in south*
*central China, Contact Dr. Xiaobin Xu at the China Meteorologi- cal Administration:*
*xiaobin_xu@189.cn*

**Response**. We have obtained the Hong Kong site observations from Prof. Tao Wang from the above
list. And we have also used the TOAR dataset to validate our model. We find that the OMI inferred
ozone trends are fairly consistent with these long term surface records.
P1 L21. Comparison of 2005-2009 and 2013-2017 OMI data indicates that mean summer afternoon surface
ozone in southern China (including urban and rural regions) has increased by 3.5 ppb over the 8-year period
and the number of episode days per summer has increased by 2.2 (as diagnosed by an extreme value model),
fairly consistent with the few long-term surface records.
P9 L6. We compared the OMI trends in Figure 6 to the trends of MDA8 ozone and number of high-ozone
days reported by the long-term TOAR sites (Schultz et al., 2018) and our own analysis for the Hok Tsui
station in Hong Kong (Wang et al., 2009). For Lin'an, Hong Kong, and the 5 sites in Taiwan, the changes of
mean ozone concentrations from 2005-2009 to 2013-2017 are 1.1, 2.3, and -0.18±2.2 ppbv (standard deviation
among the 5 sites) as estimated from OMI, compared to 0.7, 5.6 (or 5.8 in Hok Tsui station), and -0.75±3.4
ppbv for MDA8 ozone at the TOAR sites. The changes in the number of ozone episodes per summer are 1.2,
1.9, and -0.17±0.74 days in OMI, compared to 2.1, 1.8 (or 2.1 in Hok Tsui station), and -3.5±3.9 days at the
TOAR sites. These OMI inferred trends are fairly consistent with the long-term records available from surface
sites.

[Figure]

**Changes in summertime surface ozone pollution inferred from OMI**
**(2005-2009 to 2013-2017)**

**Figure** 6. Changes in surface ozone pollution in China between 2005-2009 and 2013-2017 as
inferred from OMI afternoon observations at around 13:30 local time. (a) Change in mean summer
afternoon concentrations, obtained from the difference in the mean OMI enhancements at 850-400
hPa and applying equation (1). Also shown with symbols are observed changes in mean MDA8
ozone from in situ observations in Lin'an, Hong Kong, and Taiwan reported by TOAR (Schultz et
al., 2018). Because the TOAR observations are only reported for 2005-2014, we estimate the
changes from 2005-2009 to 2013-2017 on the basis of the reported linear trends during 2005-2014
(ppb a$^{-1}$). The change of 12-15 LT ozone at the Hok Tsui station in Hong Kong is 5.8 ppb. (b)
Change in the number of high-ozone days (> 82 ppb) per summer, calculated by applying the
probability of exceeding 82 ppb (equation 8) to the daily OMI enhancements. Also shown with
symbols are observed changes of the number of days with MDA8 ozone exceeding 80 ppb at the
TOAR sites, similarly adjusted as the change from 2005-2009 to 2013-2017. The change in the number of days with 12-15 LT ozone exceeding 82 ppbv at the Hok Tsui station in Hong Kong is
2.1 days.

*Ozone at the surface and in the mid-troposphere varies greatly with transport path- way and abrupt*
*changes in air masses, and recent studies have shown that ozone in China varies with meteorology*
*[Pu et al., 2017; Zhao et al., 2018]. The authors are aware of this phenomenon as their previous*
*work has explored the impact of climate variability on ozone. Therefore I'm surprised that the*
*authors didn't first explore how surface ozone across China varies with meteorology, such as surface*
*temperatures (or temperature at 850 hpa) [Pusede et al., 2015], or with the height of the 500 hPa*
*surface [Reddy et al., 2016], both of which correlate quite well with surface ozone. The authors*
*should first determine the correlation between surface ozone and meteorology, and then compare*
*these results to what they find from OMI ozone. Does OMI give more information on surface ozone*
*than basic meteorology? Given that reanalysis data are available for all of China under all weather*
*conditions (no cloud screening) I would think that the meteorology would perform better than OMI.*
*If OMI performs less well than meteorology, is there any reason to use OMI to try to predict surface*
*ozone, when meteorological analyses are available everywhere and at all times?*

**Response**. Our focus is to evaluate the OMI observation capability, not to analyze the correlation of
ozone with meteorological variables which has been done before and would not capture the
variability in ozone driven by emissions. The reviewer makes a good point that mid-tropospheric and
surface ozone may respond similarly to meteorological conditions, which may in turn contribute to
the correlation of OMI with surface ozone. We now address this point in Section 4 by analysis of the
ozonesonde data (P6 L15-25 with new Figure S4). We also mention this in other parts of the
manuscript.
P5L18. The correlation of OMI with the MEE surface ozone data likely does not reflect a direct sensitivity of
OMI to surface ozone, which is very weak, but rather a sensitivity to boundary layer ozone extending up to a
certain depth and correlated with surface ozone.
P9L25. To better understand the correlation of OMI with surface ozone we examined vertical ozone profiles
from Hong Kong and other ozonesondes, and from the IAGOS commercial aircraft program. Some of the
correlation is driven by similar meteorology influencing ozone in the mid-troposphere (where OMI sensitivity
is maximum) and the boundary layer, but most of the correlation is driven by direct sensitivity to the boundary
layer.

*Another necessary analysis is to see if in situ observations of ozone in the mid- troposphere are*
*correlated with surface ozone. I realize that the authors did look at ozonesonde profiles above Hong*
*Kong, but they are not very frequent and they don't tell us anything about ozone in other parts of*
*China, especially in the highly polluted North China Plain. The IAGOS program has hundreds of*
*profiles above East Asia since 1995. As shown by Ding et al. [2005] and by Gaudel et al. [2018]*
*ozone in sum- mertime in the boundary layer is much greater than ozone in the mid-troposphere. The*
*difference is due to very strong ozone production in the boundary layer, versus distant source regions*
*for ozone in the mid-troposphere. If the authors conducted a trans- port study for ozone in the mid-*
*troposphere they would find that very little of the air in this layer comes from the surface of China.*
*Probably 80-90% of the mid-tropospheric above China air has either been in the mid-troposphere*

*for days, or it comes from the boundary layer far upwind of China. The authors can freely access*
*hundreds of com- mercial aircraft profiles of ozone and carbon monoxide above mainland China,*
*Hong Kong, Taiwan and South Korea from the IAGOS database. They can then apply the OMI*
*averaging kernel to the profiles and determine the relationship between IAGOS ozone in the mid- and*
*lower troposphere to ozone at the surface. Does IAGOS ozone in the mid-troposphere correlate with*
*ozone at the surface? Is the correlation any better than when surface ozone is correlated with*
*meteorology? Then compare the IAGOS relationship to the OMI relationship. Does OMI perform*
*any better than IAGOS?*
**Response.** We only find 54 profiles that can be used to validate the OMI data. This is because the
OMI crossing time is 13:30 local time and we have to use observations close to this time window.
We have added more discussion in the text.
P6 L9. We applied the same daily correlation analysis to the other ozonesonde datasets and IAGOS aircraft
measurements during 2005-2017 summers. For the 54 IAGOS vertical profiles coincident with OMI
observations, the correlation coefficient of the 950 hPa in situ ozone and 850-400 hPa OMI ozone is $R = 0.59$
($p < 0.05$) (Figure S2).

[Figure]

**Figure S2**. (a) Location and the number of tropospheric profiles at each airport in IAGOS that are
coincident with OMI retrievals. We only select these profiles between 12-15 local time. (b) Ozone
profiles from IAGOS, but mapped to OMI layers. The missing data at L21 is in part because we
only select pixels that are within 200 km from the airport along the flight path. (c) OMI ozone
profiles coincident with the ozonesondes. The correlations of unsmoothed 950 hPa ozone data in
IAGOS with the OMI retrievals for different levels are shown inset. The correlation with 850-400
hPa OMI ozone is 0.59 ($p < 0.05$)

*Figure 5 shows surface ozone trends across China which were derived from the OMI ozone product.*
*The strongest trends are in the far north of China and in the far south of China. Based on the summer*
*OMI trends (2005-2015) reported by the Tropospheric Ozone Assessment Report in supplementary*
*Figure S-24 of Gaudel et al. [2018], OMI has a strong trend across southern China but no trend*
*across northern China. Therefore I don't understand how Figure 5 can show trends across northern*
*China. It would be helpful to include a map that shows the OMI trends across China.*
**Response.** Thanks for the careful reading. The authors who plotted Figure S24 in the TOAR report
(Gaudel et al., 2018) are also coauthors of this work. The difference of trends in northern China
arises from these reasons. First, we use the 850-400 hPa ozone but Gaudel et al. (2018) uses the
tropospheric column ozone. Second, we use different methods to remove the background. Third, we
have removed the low quality L2 data but Gaudel et al. (2018) kept all of them. Now we have this
new figure in the supplement.

[Figure]

**Figure S5**. Difference of the mean OMI enhancements at 850-400 hPa from 2005-2009 to 2013-
2017 after correcting the Pacific background. Data are only shown for regions with DOFS below
400 hPa (Figure 1a) greater than 0.30.

**Referee #2**

*This is a nice study that explores the potential of OMI observations of tropospheric ozone to detect the ozone pollution over China. While it's unrealistic to use OMI data to capture the day-to-day variability of ozone pollution, the authors show extreme ozone pollution may be detectable by aggregating long-term observations using statistical methods. Overall I think this is an important study to the field, which opens up the possibility to use satellite observations to detect surface ozone pollution, but I think the authors overpromise the value of satellite data. I have several major concerns:*

**Response**. Thanks for raising these good points. This feedback has significantly improved the manuscript. Now we have a new Figure 4 showing that OMI 850-400 retrievals have limited skill in predicting the daily ozone variability in the north and we only predict the trends of ozone pollution in southern China (south of 34°N). We have new in-situ observations to validate the trends inferred from the OMI, which are shown in Figure 6.

1. *My major concern is that the authors seem to overpromise the value of OMI data for characterizing the spatial and temporal trend of ground-level ozone. The title and the abstract leave me an impression that OMI satellite data can capture the spatial distribution and the long-term trend in ground-level ozone, but the results only suggest OMI may be able to detect high ozone pollution and capture the large-scale or latitudinal variations. I suggest the authors consider revising the title, otherwise it'd be misleading to readers. The authors need to be more careful with the wording. I think this work would actually be much more valuable if the authors can clarify the limitations of OMI data, which will also be useful for preparation of next-generation satellites.*

**Response**. Thanks for making such a good point. Now we revised the title and also discussed the limitations in many parts of the main text.

[revised manuscript text omitted]

*3. Figure 5: While OMI data may be able to detect the sign of the change in ground- level ozone, the*
*magnitude of the change is less convincing to me. The authors suggest a 0.67 ppb /year increase in*
*mean ozone over China, which seems to be lower than previous studies. The point process model is*
*trained with ground-based observations in 2013-2017, but it's unknown how the model performs for*
*early years 2005 - 2009. I'd suggest the authors use available long-term ground-based ozone*
*observations to verify the long-term change. I understand long-term ground-based observations are*
*not generally available over China, but since the OMI data are global, it's possible to extend the*
*analysis to wider regions (e.g. Hong Kong, Japan) where long-term sites are available for*
*evaluation.*
**Response**. Thanks. We have new in-situ observations from TOAR and also from a Hong Kong site to
validate the trends inferred from the OMI, which are shown in Figure 6. We find the OMI inferred
trends are fairly consistent with the long-term records available from surface sites. We also add discussion
in the main text.
P4 L11. For evaluating the long-term surface ozone trends inferred from OMI, we use 2005-2014 trend
statistics for maximum daily 8-hour average (MDA8) ozone from the Tropospheric Ozone Assessment Report
(TOAR) (Schultz et al., 2018). We also have 2005-2017 JJA 12-15 LT mean ozone at the Hok Tsui station in
Hong Kong (Wang et al., 2009).
P9 L5. We compared the OMI trends in Figure 6 to the trends of MDA8 ozone and number of high-ozone days
reported by the long-term TOAR sites (Schultz et al., 2018) and our own analysis for the Hok Tsui station in
Hong Kong (Wang et al., 2009). For Lin'an, Hong Kong, and the 5 sites in Taiwan, the changes of mean
ozone concentrations from 2005-2009 to 2013-2017 are 1.1, 2.3, and -0.18±2.2 ppbv (standard deviation
among the 5 sites) as estimated from OMI, compared to 0.7, 5.6 (or 5.8 in Hok Tsui station), and -0.75±3.4
ppbv for MDA8 ozone at the TOAR sites. The changes in the number of ozone episodes per summer are 1.2,
1.9, and -0.17±0.74 days in OMI, compared to 2.1, 1.8 (or 2.1 in Hok Tsui station), and -3.5±3.9 days at the
TOAR sites. These OMI inferred trends are fairly consistent with the long-term records available from surface
sites.

**Referee 3**

*This paper explored the capability of OMI ozone columns to represent the surface O3. I feel the*
*satellite data is over-interpreted based on the evidence provided in the paper. However, I do believe*
*it will be big news if substantial improvements are made to prove that the conclusion is solid.*

**Response**. We thank the reviewer for raising so many good points, which have significantly
improved our work. Now we have a new Figure 4 showing that OMI 850-400 retrievals have limited
skill in predicting the daily ozone variability in the north and we only predict the trends of ozone
pollution in southern China (south of 34°N). We have new in-situ observations to validate the trends
inferred from the OMI, which are shown in Figure 6. And we have revised the title.
New title. Ability of the OMI satellite instrument to observe surface ozone pollution in China:
application to 2005-2017 ozone trends

General comments:
*1. The sensitivity of OMI O3 to the lower troposphere is very low. I feel that is the reason why no*
*quantitative comparison to surface observations has so far been done. I'm wondering is there any*
*improvements that have been made to make the quantitative comparison robust? Why does not the*
*quantitative comparison work for other regions, but work for China?*
**Response.** We now explain this better in the Introduction.
P2L23. However, no quantitative comparison of the satellite data to surface observations has so far been done.
Surface ozone network data are available in the US and Europe but levels are generally too low to enable
statistically meaningful validation. Ozone levels in China are much higher (Lu et al., 2018). The high density
of the MEE network, combined with vertical profile information from ozonesondes and aircraft, provides a
unique opportunity for evaluating quantitatively the ability of OMI to observe ozone pollution.

*2. The robustness of the residual. How large is the temporal and spatial variations of the*
*background? Is it likely that such variations bring significant uncertainties to the subtraction?*
**Response**. Thanks for making this good point. We have tested different approaches to correct the
background and the results are consistent with what have presented in the paper. In the text, we make
it simple by saying this.
P4L1. We examined different spatial and temporal averaging domains for the North Pacific background and
found little effect on the residual.
The uncertainty related to the background correction can be found in these two figures.

[Figure]

**Figure SX**. Standard deviation of monthly OMI 850-400 hPa ozone during 2005-2017 summers.

[Figure]

**Figure SX**. The mean ozone enhancement (left panel), daily correlation of OMI and MEE ozone
(mid panel), and the OMI inferred changes of mean ozone concentrations from 2005-2009 to 2013-
2017 using different approaches correcting the OMI drift. In the top panel, we subtract the monthly
mean Pacific background (150°E-150°W) for the corresponding latitude and season. In the bottom
panel, we subtract the monthly mean mid-latitude ozone for the corresponding latitude and month.
3. *The correlation between MEE and OMI. The correlation seems to be related with the dependence*
*of O3 on latitude. I suggest additional analysis here to prove that is not the case.*
**Response**. We are not sure if the reviewer is referring to Figure 1f here. In Figure 1f, the correlation is higher in the south and lower in the north. This is because in the northern China, OMI 850-400 hPa
ozone has lower sensitivity in the boundary layer, more likely to be influenced by the upper
tropospheric ozone variability and stratosphere-troposphere exchange. We have added new
discussion in the text.

P6 L27. We find that the low correlation of OMI with boundary layer ozone in the northern
ozonesonde data is due not only to the low DOFS but also to a large variability of ozone in the
upper troposphere. Figure 4 (left panel) shows the standard deviation of daily OMI 400-200 hPa
ozone during 2005-2017 summers, indicating that upper tropospheric ozone has much higher
variability in the north (> 34°N) than in the south. This is related to the location of the jet stream
and more active stratospheric influence (Hayashida et al., 2015). Figure 4 (right panel) displays the
vertical profiles of ozone standard deviations for the five ozonesonde sites. For the two sites north
of 34°N, the ozone variability becomes very large above 8 km. Since the OMI 850-400 hPa
retrieval also contains information from above 400 hPa, this upper tropospheric variability causes a
large amount of noise that masks the signal from boundary layer variability. For the three sites
south of 34°N, the ozone variability in the boundary layer is much higher than in the free
troposphere and the upper tropospheric ozone variability still remains low even above 8 km. In the
rest of this paper we focus our attention on ozone episodes and the long-term trends in southern
China (south of 34°N).

If the reviewer refers to Figure 1d, we now make it clear that we have corrected the background that
is dependent on latitudes.
P3 L23. To remove this gradient and also any long-term uniform drift in the data, we subtract the monthly
mean Pacific background (150$^o$E-150$^o$W) for the corresponding latitude and month
P4 L22. After subtracting the North Pacific background for the corresponding latitude in month, we obtain the
OMI ozone enhancements shown in Figure 1d.
P4 L23. The spatial correlation coefficient between the OMI ozone enhancements and the MEE surface
network is $R$ = 0.73 over eastern China. The correlation is driven in part by the latitudinal gradient but also by
the enhancements in the large megacity clusters identified as rectangles in Figure 1b. Thus the correlation
coefficient is $R$ = 0.55 for the 26-34°N latitude band including YRD, SCB, and Wuhan.
*Specific comments:*
*1. "We exclude outliers with over 35 Dobson Units (DU) at 850- 400 hPa (>99th percentile in*
*eastern China) and exclude July 2011 when the retrievals are anomalously high." Please give the*
*reference to the exclusion. Otherwise, please quantify the influence of the exclusion.*
**Response**. Thanks for pointing this out. We delete this because we don't use the July 2011 data and
not excluding the extremely high data has little effect on our result.
*2. "We see that high-ozone episodes in the 950-850 hPa sonde data are systematically associated*
*with high OMI values, though the converse does not always hold." Additional explanation for the*
*reason is expected.*
**Response**. Thanks. Now we say
P6L4. We see that high-ozone episodes in the 950-850 hPa sonde data are systematically associated with high
OMI values, though the converse does not always hold because free tropospheric enhancements affecting OMI

can also occur.

---

## Author Response (AR2)

**Response to referee comments on "An evaluation of the OMI satellite instrument's ability to**
**observe boundary layer ozone pollution across China: application to 2005-2017 ozone trends"**
We thank the referees for their careful reading of the manuscript and the valuable comments. This
document is organized as follows: the Referee's comments are in *italic*, our responses are in plain
text, and all the revisions in the manuscript are shown in blue. The line numbers in this document
refer to the updated manuscript.

**Referee 1**
*The authors have conducted a thorough revision of their manuscript and I appreciate their efforts*
*to evaluate OMI against IAGOS profiles, additional ozonesondes and long-term surface*
*observations. The results are much more defensible, especially since the authors now focus on*
*southern China and have clarified that their analysis is representative of the boundary layer rather*
*than the surface. The new Figure 4 is also very helpful as it shows that ozone variability in the*
*boundary layer is greater than in the mid-troposphere (in southern China), which can help to*
*distinguish boundary layer pollution episodes. I still have a few concerns, as described below, and I*
*could recommend the paper for publication if the authors can address these concerns.*
**Response**. We thank the reviewer for this detailed feedback, which has helped improved our work.
*The paper has been revised to address the concerns of the referees that OMI does not detect*
*surface ozone but rather has some sensitivity to ozone in the deeper boundary layer, provided that*
*the enhancements are relatively strong compared to the mid-troposphere. However, this revised*
*focus is incomplete as the paper still states or implies, in many places throughout the manuscript,*
*that OMI is detecting surface ozone. The most important statement in the revision is this sentence at*
*the beginning of section 4:*
*"The correlation of OMI with the MEE surface ozone data likely does not reflect a direct sensitivity*
*of OMI to surface ozone, which is very weak, but rather a sensitivity to boundary layer ozone*
*extending up to a certain depth and correlated with surface ozone."*
*The reader should not have to wait until the middle of the paper to read this very important*
*statement. It needs to appear in the Introduction and then the authors need to thoroughly revise*
*their paper to make sure that the discussion and the conclusions are consistent with this statement.*
**Response**. Thanks for pointing this out. We have revised a lot of sentences to make sure the
statements are consistent. We also have this sentence in the introduction.
P2L22. Even if sensitivity to the boundary layer is low, the enhancements can be sufficiently large
to enable detection.
*For example, the authors changed the title of the paper so that it reflects the analysis in the revised*
*manuscript, but it's still not quite right. In their response to the comments of all three referees the*
*authors clarified that the OMI retrievals are reflective of the boundary layer rather than the*
*surface. But the title gives the impression that OMI has the ability to observe surface ozone*
*pollution. At a minimum the title needs to be revised, with "surface" replaced with "boundary*
*layer". But even then the reader has the impression that the product can be used for all of China,*
*whereas the product is not effective in northern China. A better title would be something like:*
*"An evaluation of the OMI satellite instrument's ability to observe boundary layer ozone pollution*
*across China: application to 2005-2017 ozone trends"*
**Response**. We have revised the title.
New title. An evaluation of the OMI satellite instrument's ability to observe boundary layer ozone
pollution across China: application to 2005-2017 ozone trends
*Other statements in the paper that are not consistent with the first sentence at the beginning of*
*Section 4 are:*
*1) First sentence of the abstract: "Nadir-viewing satellite observations of tropospheric ozone in the*
*UV have been shown to detect surface ozone pollution episodes but no quantitative evaluation of*

*this ability has been done so far." This isn't entirely accurate. A better summary statement would*
*be "...have been shown to have some sensitively to surface or near-surface ozone pollution*
*episodes"*
**Response**. We have revised accordingly.
*2) Second sentence of the abstract: "Here we use 2013-2017 surface ozone data from the new*
*China Ministry of Ecology and Environment (MEE) network of ~1000 sites, together with vertical*
*profiles from ozonesondes and aircraft, to quantify the ability of OMI tropospheric ozone retrievals*
*to characterize surface ozone pollution in China."*
*This also gives the impression that OMI can detect surface ozone, when you should be talking*
*about the correlation between OMI and surface ozone. A better statement would be:*
*"...to quantify the correlation between OMI tropospheric ozone retrievals and surface ozone*
*pollution events in China."*
**Response**. We have revised the sentence.
*3) Third sentence of the abstract:*
*"After subtracting the Pacific background, the 2013-2017 mean OMI ozone enhancements over*
*eastern China can quantify the spatial distribution of mean summer afternoon surface ozone with a*
*precision of 8.4 ppb and a spatial correlation coefficient R=0.73." This sentence would be more*
*accurate as:*
*"After subtracting the Pacific background, the correlation between observed surface ozone and the*
*2013-2017 mean OMI ozone enhancements over eastern China can be used to estimate the spatial*
*distribution of mean summer afternoon surface ozone with a precision of 8.4 ppb and a spatial*
*correlation coefficient R=0.73."*
**Response**. We have revised the sentence.
*4) Fifth sentence of the abstract:*
*"OMI is much more successful at capturing the day-to-day variability of surface ozone at sites in*
*southern China <34 N (R = 0.3-0.6) than in northern China (R = 0.1-0.3) because of weaker*
*retrieval sensitivity and larger upper tropospheric variability in the north."*
*Would be better as:*
*"In terms of day-to-day variability, OMI has a higher correlation with observed surface ozone at*
*sites in southern China <34 N (R = 0.3-0.6) than in northern China (R = 0.1-0.3) because of*
*weaker retrieval sensitivity and larger upper tropospheric variability in the north."*
**Response**. We have revised the sentence.
*There are many more statements like this throughout the paper that need to be revised, following*
*the examples above.*
*In Section 2 the following important statement appears:*
*"We focus on summer when ozone pollution in China is most severe and when OMI has the*
*strongest sensitivity"*
*A similar statement needs to appear in the abstract and introduction so that the reader is aware*
*that the evaluation has so far only focused on a region and time of year when boundary layer ozone*
*is very high.*
**Response**. Now we say this in the abstract.

P1L16. We focus on summer when ozone pollution in China is most severe and when OMI has the
strongest sensitivity.
*In Section 3 you provide the following equation and state that "With such a precision, OMI can*
*provide useful information on mean summer afternoon levels of surface ozone in polluted regions.":*
*[O3] = 6.9 ΔΩ + 24.6 ± 8.4*
*This presentation (and in the abstract as well) gives the reader the impression that they can take*
*this equation and an OMI retrieval above a given location on a given day, and estimate the*
*afternoon ozone on that day. For example, an OMI enhancement of 5 DU would indicate a surface*
*ozone mixing ratio of 59.1 ± 8.4 ppbv, or a range of 50.7 to 67.5 ppbv. This sounds quite good, but*
*when I look at Figure 1e, I can see that the full range of surface ozone values for an enhancement*
*of 5 DU is 35-80 ppbv. And of course, this equation is not derived for estimating ozone on a given*
*day, it is derived from data averaged over 5 years. Therefore the text needs to clearly state that the*
*equation can only provide an estimate of multi-year average ozone. Accordingly, the text in the*
*abstract also needs to be modified to more accurately represent the limitations of the equation. The*
*abstract should provide a statement similar to:*
*"After subtracting the Pacific background, the correlation between observed surface ozone and the*
*2013-2017 mean OMI ozone enhancements over eastern China can be used to estimate the broad*
*multi-year spatial distribution of mean summer afternoon surface ozone with a precision of 8 ppb*
*and a spatial correlation coefficient R=0.73." Note that I emphasized "multi-year", and changed*
*8.4 to 8 (8.4 is far too precise for such a rough estimate).*
**Response**. Thanks. We have revised it in the abstract and also in these places.
P4L27. From there one can estimate **multi-year average** surface ozone (ppb) on the basis of the
observed OMI enhancement (DU) as...
P9L17. After subtracting the contribution from the North Pacific background, we find that the OMI
enhancement over eastern China can reproduce the observed spatial distribution of **multi-year**
mean summer afternoon ozone concentrations at the MEE sites, with a correlation coefficient $R =$
0.73 and a precision of 8 ppb.
*Section 4*
*Here you state: "The Hong Kong ozonesonde data thus indicate that OMI can quantify the*
*frequency of high-ozone episodes in the boundary layer even if it may not be reliable for individual*
*events."*
*This is a good summary of OMI's capability, and it's this type of statement that your paper should*
*focus on, rather than emphasizing the equation in Section 3.*
**Response**. We have it in the conclusion now.
P9L27. The Hong Kong ozonesonde data also indicates that OMI can quantify the frequency of
high-ozone episodes in the boundary layer even if it may not be reliable for individual events.
*In the last paragraph of the Introduction you state:*
*"However, no quantitative comparison of the satellite data to surface observations has so far been*
*done." The implication of this statement is that no one has ever bothered to compare satellite*
*retrievals to surface ozone, when it could be done very easily. But the reason this has not been done*

*before is because the available tropospheric column ozone products don't have a true sensitivity to*
*surface ozone. A better statement would be:*
*"While the available tropospheric column ozone products don't have a true sensitivity to surface*
*ozone, here we demonstrate that the strong ozone enhancements in the summertime boundary layer*
*of southern China can be used to derive a relationship between OMI tropospheric ozone and*
*observed surface ozone "*
*Also, the above statement discounts the important advances made by the scientists who are*
*combining IASI and GOME-2 to produce retrievals of tropospheric ozone between the surface and*
*3 km. Figure 8 and Figure 9 of Gaudel et al. [2018] show tropospheric column ozone between the*
*surface and 3 km above China and Europe, based on retrievals from the IASI and GOME-2*
*instruments. These retrievals compared well to IAGOS ozone profiles above China (see Figure S8*
*in the supplement to Gaudel et al. 2018). (Please also see Dufour et al [2018] who look at ozone*
*across China from the surface to 6 km using IASI). These papers should be referenced in the*
*Introduction.*
**Response**. We have added these two references. About the ability of OMI to detect surface ozone
pollution, we think it is better to briefly mention it in the introduction and then give more details in
later discussion. Now we say this in the introduction.
P2L23. Even if sensitivity to the boundary layer is low, the enhancements can be sufficiently large
to enable detection.
*The following sentence appears in the Introduction:*
*"Retrieval of tropospheric ozone (only ~10% of the column) from these instruments has mostly*
*been done in the past by subtracting independent satellite measurements of stratospheric ozone*
*(Fishman et al., 1987; Ziemke et al., 2011)."*
*Another common method that should be mentioned is the cloud-slicing technique. This technique*
*was recently used in an excellent paper by Ziemke et al. [2019] who show 40 years of ozone*
*increases above southern Asia from the TOMS and OMI instruments.*
**Response**. Now we say.
P2L14. Retrieval of tropospheric ozone (only ~10% of the column) from these instruments has
mostly been done in the past by subtracting independent satellite measurements of stratospheric
ozone (Fishman et al., 1987; Ziemke et al., 2011) or using the convective cloud differential method
(Ziemke et al., 1998, 2019).
*The IASI+GOME2 results for China [Gaudel et al., 2018] show that high ozone in spring and*
*summer can extend over the North Pacific Ocean. Why do you not show OMI data above the ocean*
*in Figure 6 and Figure S5? However, you do show ozone above Taiwan in Figure 6 and compare*
*OMI to the TOAR observations from Taiwan, even though you don't use Taiwanese surface*
*observations in your equation 8.*
**Response**. Thanks. Now we say this in text.
P8L22. Here we have extended the trend analysis to Taiwan because of the opportunity to compare
to surface records.
*I don't understand Figure 5b. Figure 5a shows some clear differences between the observed and*
*predicted high ozone episodes for a threshold of 82 ppbv. Yet Figure 5b with higher thresholds*
*shows nearly perfect agreement between the observed and predicted. Is this for all sites averaged*

*together? What is the scatter for the individual sites?*
**Response**. Now we make it clear the Figure 5b uses all data in eastern China south of 34°N. We
also reported the spatial correlation of ozone episode days in surface sites and from OMI.
P8L15. However, we find that the model can also accurately estimate the probability of exceedance
above higher thresholds (Figure 5b) **for the ensemble of eastern China sites south of 34⁰N**, which
confirms the property of threshold invariance of an extreme value model (Cole, 2001).
P8L12. The results show that the predicted fraction of ozone episodes resembles that observed, with
a spatial correlation of 0.62 (Figure 5a).

Dufour, G., Eremenko, M., Beekmann, M., Cuesta, J., Foret, G., Fortems-Cheiney, A., Lachâtre, M.,
Lin, W., Liu, Y., Xu, X., and Zhang, Y.: Lower tropospheric ozone over the North China Plain:
variability and trends revealed by IASI satellite observations for 2008–2016, Atmos. Chem. Phys.,
18, 16439-16459, https://doi.org/10.5194/acp-18-16439-2018, 2018.
Gaudel, A., et al. (2018), Tropospheric Ozone Assessment Report: Present-day distribution and
trends of tropospheric ozone relevant to climate and global atmospheric chemistry model evaluation,
Elem. Sci. Anth., 6(1):39, DOI: https://doi.org/10.1525/elementa.291
Ziemke, J. R., Oman, L. D., Strode, S. A., Douglass, A. R., Olsen, M. A., McPeters, R. D., Bhartia,
P. K., Froidevaux, L., Labow, G. J., Witte, J. C., Thompson, A. M., Haffner, D. P., Kramarova, N.
A., Frith, S. M., Huang, L.-K., Jaross, G. R., Seftor, C. J., Deland, M. T., and Taylor, S. L.: Trends
in global tropospheric ozone inferred from a composite record of TOMS/OMI/MLS/OMPS satellite
measurements and the MERRA-2 GMI simulation , Atmos. Chem. Phys., 19, 3257-3269,
https://doi.org/10.5194/acp-19-3257-2019, 2019.

**Referee 2**
*The authors well addressed the reviewers' comments and concerns. It's a nice addition to clarify*
*the limitation of satellite data for detecting ozone pollution. The additional evaluation with ground-*
*based observations makes the results more robust. However, I have some concerns with the*
*evaluation with TOAR sites (Page 9, Line 10). There is large discrepancy between OMI-derived*
*and observed trends in ozone, especially at the Hok Tsui site, where OMI underestimates the*
*change by more than a factor of two. I'd suggest the authors provide possible explanations for such*
*discrepancy, instead of making the questionable statement that 'they are fairly consistent'. Given*
*the discrepancy, I think it's necessary to provide an uncertainty estimate for the derived trends.*
**Response**. Thanks. Now we have made these changes in text.
P9L8. For Lin'an, Hong Kong, and the 5 sites in Taiwan (we report the mean value here), the
changes of mean ozone concentrations from 2005-2009 to 2013-2017 are 1.1±3.6, 2.3±3.3, and -
0.18±2.9 ppbv as estimated from OMI, compared to 0.7±3.6, 5.6±3.9 (or 5.8±1.3 in Hok Tsui
station), and -0.75±2.5 ppbv for MDA8 ozone at the TOAR sites. The changes in the number of
ozone episodes per summer are 1.2±0.7, 1.9±0.24, and -0.17±0.14 days in OMI, compared to
2.1±4.4, 1.8±1.7 (or 2.1±1.1 in Hok Tsui station), and -3.5±1.8 days at the TOAR sites. The
standard errors are obtained by applying a parametric bootstrap method. The OMI inferred trends
are generally consistent with the long-term records available from surface sites.

**An evaluation of the OMI satellite instrument's ability to observe boundary layer ozone pollution across China: application to 2005-2017 ozone trends**

Lu Shen[1], Daniel J. Jacob[1], Xiong Liu[2], Guanyu Huang[3], Ke Li[1], Hong Liao[4], Tao Wang[5]

[1]John A. Paulson School of Engineering and Applied Sciences, Harvard University, Cambridge, MA 02138, USA
[2]Harvard-Smithsonian Center for Astrophysics, Cambridge, Massachusetts 02138, USA
[3]Environmental & Health Sciences, Spelman College, Atlanta, Georgia 30314, USA
[4]School of Environmental Science and Engineering, Nanjing University of Information Science & Technology, Nanjing 210044, China
[5]Department of Civil and Environmental Engineering, The Hong Kong Polytechnic University, Hong Kong

*Correspondence to*: Lu Shen (lshen@fas.harvard.edu)

**Abstract.** Nadir-viewing satellite observations of tropospheric ozone in the UV have been shown to have some sensitivity to boundary layer ozone pollution episodes but no quantitative evaluation of this ability has been done so far. Here we use 2013-2017 surface ozone data from the new China Ministry of Ecology and Environment (MEE) network of ~1000 sites, together with vertical profiles from ozonesondes and aircraft, to quantify the ability of tropospheric ozone retrievals from the OMI satellite instrument and to detect boundary layer ozone pollution in China. We focus on summer when ozone pollution in China is most severe and when OMI has the strongest sensitivity. After subtracting the Pacific background, we find that the 2013-2017 mean OMI ozone enhancements over eastern China have strong spatial correlation with the corresponding multiyear means in the surface afternoon observations ($R = 0.73$), and that OMI can estimate these multiyear means in summer afternoon surface ozone with a precision of 8 ppb. The OMI data show significantly higher values on observed surface ozone episode days (>82 ppb) than on non-episode days. Day-to-day correlations with surface ozone are much weaker due to OMI instrument noise, and are stronger for sites in southern China (<34°N; $R = 0.3$-$0.6$) than in northern China ($R = 0.1$-$0.3$) because of weaker retrieval sensitivity and larger upper tropospheric variability in the north. Ozonesonde data show that much of the variability of OMI ozone over southern China in summer is driven by the boundary layer. Comparison of 2005-2009 and 2013-2017 OMI data indicates that mean summer afternoon surface ozone in southern China (including urban and rural regions) has increased by $3.5\pm3.0$ ppb over the 8-year period and that the number of episode days per summer has increased by $2.2\pm0.4$ (as diagnosed by an extreme value model), generally consistent with the few long-term surface records. Ozone increases have been particularly large in the Yangtze River Delta and in the Hubei, Guangxi, and Hainan provinces.

Lu Shen 4/18/19 8:00 AM

Jacob, Daniel J. 4/20/19 3:08 PM

Jacob, Daniel J. 4/20/19 3:10 PM

Lu Shen 4/18/19 8:07 AM

Jacob, Daniel J. 4/20/19 3:11 PM

Jacob, Daniel J. 4/20/19 3:14 PM

Jacob, Daniel J. 4/20/19 3:16 PM

Lu Shen 4/18/19 8:16 AM

Jacob, Daniel J. 4/20/19 3:17 PM

Jacob, Daniel J. 4/20/19 3:19 PM

Jacob, Daniel J. 4/20/19 3:19 PM

Jacob, Daniel J. 4/20/19 3:49 PM

Jacob, Daniel J. 4/20/19 3:49 PM

Jacob, Daniel J. 4/20/19 8:20 PM

[revised manuscript text omitted]